# Robust encoding of natural stimuli by neuronal response sequences in monkey visual cortex

Yang Yiling[1,2,3], Katharine Shapcott [1,4], Alina Peter[1,2,3], Johanna Klon-Lipok[5], Huang Xuhui[6,7], Andreea Lazar[1] & Wolf Singer [1,4,5] ✉

Parallel multisite recordings in the visual cortex of trained monkeys revealed that the responses of spatially distributed neurons to natural scenes are ordered in sequences. The rank order of these sequences is stimulus-specific and maintained even if the absolute timing of the responses is modified by manipulating stimulus parameters. The stimulus specificity of these sequences was highest when they were evoked by natural stimuli and deteriorated for stimulus versions in which certain statistical regularities were removed. This suggests that the response sequences result from a matching operation between sensory evidence and priors stored in the cortical network. Decoders trained on sequence order performed as well as decoders trained on rate vectors but the former could decode stimulus identity from considerably shorter response intervals than the latter. A simulated recurrent network reproduced similarly structured stimulus-specific response sequences, particularly once it was familiarized with the stimuli through non-supervised Hebbian learning. We propose that recurrent processing transforms signals from stationary visual scenes into sequential responses whose rank order is the result of a Bayesian matching operation. If this temporal code were used by the visual system it would allow for ultrafast processing of visual scenes.

Neurons convey information through the frequency of their discharges (rate code) and the fine temporal structure of their discharges (temporal code). While there is consensus about the importance of temporal codes in auditory and motor processing it is less clear whether temporal codes play a role in visual pattern recognition. Artificial pattern recognition systems based on deep neural networks (DNN) share numerous similarities with the organization and performance of visual processing architectures of the mammalian visual system. The functional properties of nodes in deep DNNs trained to classify visual objects resemble closely the response properties of neurons at corresponding levels of the processing hierarchy in the mammalian visual system[1–3]. This supports the notion that essential visual functions are realized by serial processing in feedforward architectures. However, temporal codes play no or only a very limited role in these artificial systems because their predominantly feedforward connectivity is not well suited to support temporal dynamics. Biological nervous systems, by contrast, exhibit highly complex dynamics due to the abundant recurrent interactions mediated by both recurrent connectivity within processing layers and feedback loops from higher to lower processing levels (for review see Singer[4]). Especially in supragranular but to some extent also in infragranular layers of cortical areas, neurons are coupled reciprocally by recurrent connections that run tangentially to the cortical surface[5–7].

[1]Ernst Strüngmann Institute (ESI) for Neuroscience in Cooperation with Max Planck Society, 60528 Frankfurt am Main, Germany. [2]International Max Planck Research School (IMPRS) for Neural Circuits, 60438 Frankfurt am Main, Germany. [3]Faculty of Biological Sciences, Goethe-University Frankfurt am Main, 60438 Frankfurt am Main, Germany. [4]Frankfurt Institute for Advanced Studies, 60438 Frankfurt am Main, Germany. [5]Max Planck Institute for Brain Research, 60438 Frankfurt am Main, Germany. [6]Intelligent Science and Technology Academy, China Aerospace Science and Industry Corporation (CASIC), 100144 Beijing, China. [7]Institute of Automation, Chinese Academy of Sciences, 100190 Beijing, China. ✉e-mail: wolf.singer@brain.mpg.de

These intra-areal recurrent connections are complemented by equally abundant recurrent connections from higher to lower areas[8,9].

The dynamics of recurrent neural networks (RNNs) are exploited for computations in artificial RNNs[10–17] but it is less clear to which extent biological RNNs capitalize on their dynamics to achieve specific functions (for review, see Singer[4], Muller et al.[18]). One characteristic feature of recurrent networks is that they respond to perturbations with sequential activation of their nodes[19–22]. The sequence order of these sequential responses depends on the functional architecture of the RNNs[13,23] and the structure of the perturbation. Developmental studies have revealed that the recurrent connections in the mammalian visual cortex that link feature selective nodes or columns are shaped by visual experience. Connections between nodes responding to features that frequently co-occur in natural environments get strengthened[6,24–31]. Hence the functional architecture of the RNNs in the visual cortex contains information about the statistical regularities of natural scenes; and there is evidence that the information stored in the synaptic weights of the recurrent connections serves as prior for the processing of visual information ([15,17]; for review see Singer[4]). Taken together these premises predict that stationary flashed visual stimuli should cause a sequential activation of neurons in the visual cortex and that the resulting sequences should reflect the match between sensory evidence and the priors stored in the functional architecture of the RNN. If the order of sequentially activated neurons carried stimulus-specific information, this temporal code could complement the information carried by rate codes. As pointed out by Thorpe et al. (2001) and van Rullen et al. (2001) such a temporal code would be more compatible with processing speed than a rate code because the rank order of sequential responses can be decoded much faster than the amplitude of discharge rates[32,33].

In order to search for response sequences we recorded neuronal activity with implanted microelectrode arrays from V4 and V1 of monkeys trained on a fixation task and evoked responses with natural scene stimuli. We reasoned that the hypothetical response sequences, if they participated in coding, would have to occur within the very early transient response because most of the stimulus-specific information is contained in this response phase[34–46]. As we anticipated that it might be difficult to detect response sequences in this transient but information-rich response segment, we therefore expanded the response transient by slowly ramping up the intensity of the stimuli rather than flashing them as is common in visual experiments. This approach revealed stimulus-specific response sequences. Therefore, we examined in addition to which extent the sequences reflected the quality of the match between sensory evidence and priors by presenting in addition to the natural scenes manipulated versions in which we had removed in a graded way some of the regularities characterizing natural scenes.

In order to also assess the amount of rate-coded information we investigated in a time-resolved manner how much stimulus-specific information is contained in the rate vectors of the population discharges and how the information retrievable from these rate vectors is influenced by the match between sensory evidence and stored priors. In most cortical neurons the initial transient response is followed by sustained low-frequency discharges. We found that stimulus-specific information was also contained in these sustained responses and persisted over longer periods for natural than manipulated stimuli. This further supports the notion that intracortical interactions mediated by intra-areal recurrent connections[15,47,48] and/or feedback from higher cortical areas[49–52] shape responses to sensory input and enhance decodability.

## Results

### Ramping stimulus intensity unveils stimulus-specific response sequences

We developed an experimental paradigm that allowed us to slow down the transient component of responses to visual stimuli and to resolve the fine temporal structure of population responses. Rather than presenting stimuli with sudden onset, we gradually ramped up stimulus intensity (opacity, or alpha value, Fig. 1a) such that the stimulus emerged slowly from the blank background ("Methods"). Stimuli consisted of natural scenes and their manipulated versions, in which certain features were eliminated (see "Methods" and sections below). These stimuli were presented with different onset time courses: no ramp (i.e., step increase to full intensity), fast ramp (500 or 600 ms to full intensity), and slow ramp (1000 or 1200 ms to full intensity). Subsequently, stimulus intensity was held constant for at least another 500 ms. We used two levels for maximal intensity (alpha = 10% or 30%, for low and high-intensity conditions, respectively). Four awake macaque monkeys were presented with these stimuli in a passive viewing task and multi-unit activity (MUA) was recorded from visual area V4 with a 64-channel Utah array (Blackrock Microsystem, Salt Lake City, Utah, USA. Supplementary Fig. 1) in two monkeys and with a 32-channel Microdrive (Gray Matter Research, Bozeman, Montana, USA) from V1 in the other two monkeys. We did not sort for single units and analyzed only MUA in this study. We present the findings obtained from area V4 in the main text and refer readers to the supplementary information the results from area V1.

The ramping stimuli reduced the peak amplitudes of the transient response components (no ramp $51.57 \pm 0.44$ spikes/s, s.e.m.; fast ramp $48.59 \pm 0.45$ spikes/s; slow ramp $45.11 \pm 0.45$ spikes/s, corresponding to reductions of 5.8% and 12.5%, respectively; all stimulus and intensity conditions combined, $F_{2,927} = 52.79$, $p < 0.01$, one-way ANOVA; all pairwise comparisons $p < 0.01$). Ramping also increased the peak latencies of the population firing rates. When summed across electrodes and averaged over repeated trials, the median latencies were 176 ms for the no ramp, 270 ms for the fast ramp, and 358 ms for the slow ramp, corresponding to an increase of 53.4% and 100.3%, respectively ($X^2 = 347.06$, $p < 0.01$, Mood's median test; all pair-wise comparisons $p < 0.01$, Wilcoxon rank-sum test. Figure 1b). We then determined the latency of the firing rate peak for each channel ("Methods") in the slow ramp condition and rank-ordered the channels according to the latencies of responses for a given stimulus. As shown in Fig. 1c (and Supplementary Fig. 2a), each stimulus evoked a characteristic sequence of responses, the rank order of which was stimulus-specific. To test whether the sequence order was preserved between ramp conditions, we used the rank order of response latencies in the slow ramp condition to sort the responses in the fast ramp condition. This revealed that the stimulus-specific rank orders of response latencies were by and large preserved for the fast ramp condition although the absolute latencies had decreased (diagonal panels in Fig. 1d and Supplementary Fig. 2b). In the no ramp condition, these response sequences were no longer resolvable by visual inspection (Supplementary Fig. 3). To quantify the similarity between sequence orders induced by the same stimulus, we calculated the Spearman's rank order correlations between the sequences obtained from slow and fast ramp conditions for each stimulus. These correlations were statistically significant ($p < 0.05$, diagonal panels in Fig. 1d and Supplementary Fig. 2b), confirming that the rank order was independent of the time course of stimulus presentation. The same results were obtained when the onset rather than the peak latencies of the responses were evaluated (Supplementary Fig. 4).

Importantly, there were no significant correlations between the sequences evoked by different stimuli (off-diagonal panels in Fig. 1d and Supplementary Fig. 2b). This was true even for sequences induced at the same ramp conditions where stimulus onset kinetics were identical for different stimuli (Supplementary Fig. 3b, d). This lack of correlation confirms that the rank orders were stimulus-specific and differed between stimuli. Using the rank orders derived from trial-averaged response sequences as template, we could decode stimulus identity from single-trial responses irrespective of the ramp and contrast condition (Fig. 1e and Supplementary Fig. 5a). The templates

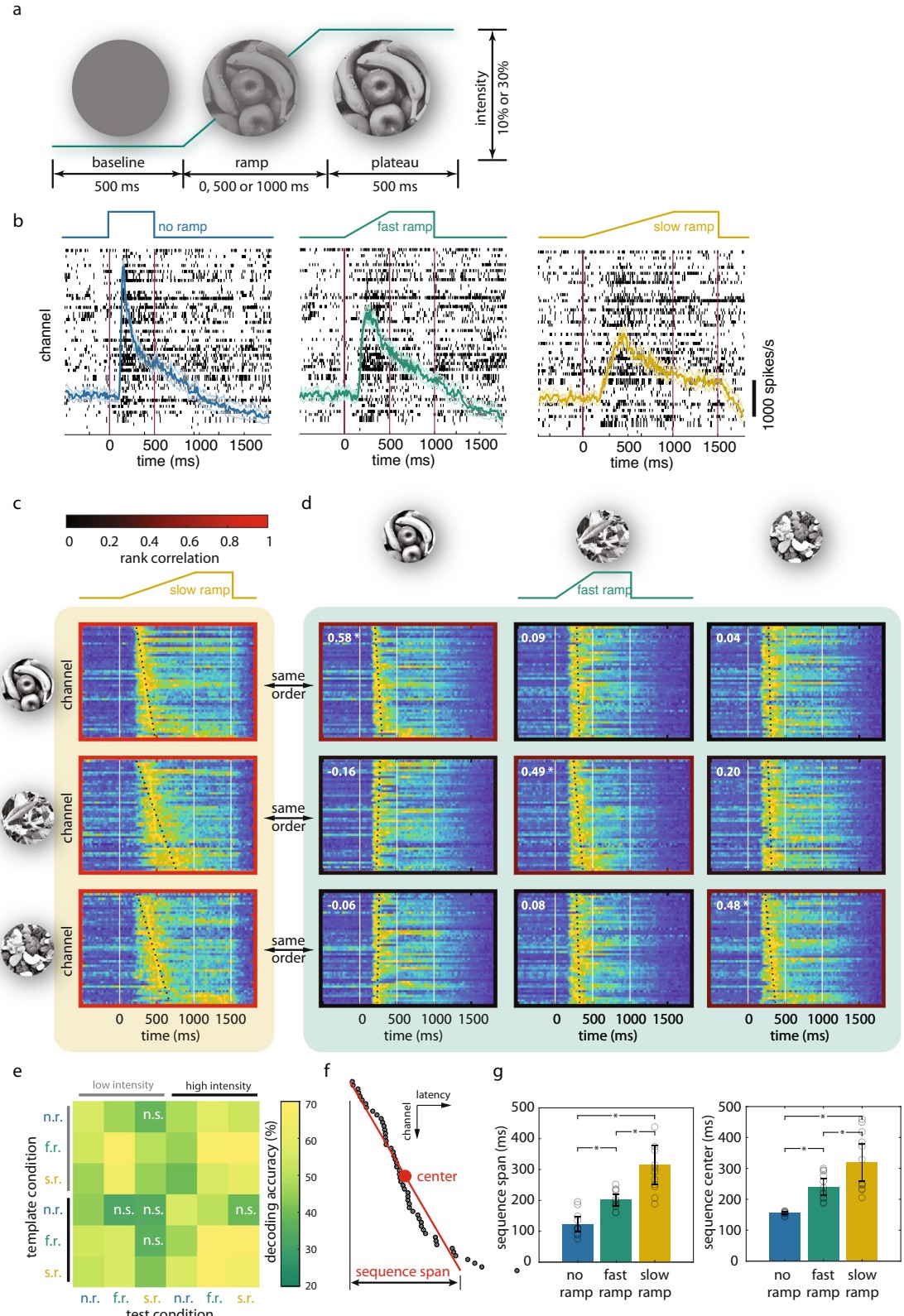

obtained from responses in the various ramp and intensity conditions predicted correctly stimulus identity from the responses evoked in the respective other conditions (Fig. 1e, pooled accuracy across conditions: 53.36 ± 1.85% (s.e.m.), $t_{35} = 28.54$, $p < 0.01$). The same results were obtained when the rank orders were derived from onset rather than peak latencies of the responses (Supplementary Fig. 5b). In other words, the "rank order decoders" generalized across conditions (ramp

duration and intensity) although the various conditions introduced substantial differences in the absolute timing of onset and peak latencies. Importantly, there was also no correlation between onset latency and firing rate amplitude of the responses (Supplementary Fig. 6). This argues against a trivial relation between onset latency and strength of the responses. Taken together, these results indicate that the stimulus-specific rank order of the sequential activation of network

**Fig. 1 | Ramping stimulus intensity reveals scalable and stimulus-specific response sequences. a** Experimental paradigm. Stimulus intensity (alpha value) increases linearly from 0 to maximal intensity (alpha = 10% or 30%). The rise times vary from 0 ms (no ramp, i.e., step onset), to 500 ms (fast ramp) and 1000 ms (slow ramp) for Monkey K (shown here). For the other monkey H see Supplementary Fig. 2. **b** Average firing rates for different ramp conditions. Raster plots of single-trial MUA responses to one of the three stimuli in the no ramp (left), fast ramp (middle), and slow ramp (right) condition. The colored traces show the average firing rate per channel for each ramp condition. Shaded areas denote 95% confidence level. **c, d** Stimulus-specificity and ramp-invariance of response sequences. **c** Firing rate responses (normalized to peak amplitude for better visualization) of the different recording channels for three different stimuli (indicated on the left) and rank-ordered according to peak latency for the slow ramp condition. **d** Comparison of the rank orders of sequences evoked by different stimuli and different ramp conditions. The responses shown in this 3-by-3 array of panels were all evoked by the fast ramp and sorted according to the rank order derived from the sequences evoked by the slow ramp in (**c**). The rows show how the sequences evoked by different stimuli (indicated above the columns) appear when sorted according to the rank order obtained in (**c**). The numbers in the panels and also the color of the frames (color scale in **c**) indicate the rank order correlation between the sequences evoked by the respective slow and fast ramps. High correlation

($*p < 0.05$) implies high consistency of rank orders. White vertical lines in each panel indicate ramp onset, ramp offset, and the end of the plateau period, respectively. Dotted lines are the best linear fits to the sequences. **e** Accuracy and generalization of decoding stimulus identity from templates of rank orders. Templates were obtained from a particular condition indicated on the ordinate (rows. n.r. = no ramp; f.r. = fast ramp; s.r. = slow ramp) and for the two intensity conditions (marked by light and dark gray bars). These templates were then used to categorize the respective test stimuli from the sequence order of the responses evoked under the respective conditions indicated on the abscissa. Decoding accuracy (%) is color-coded. n.s.: not significant ($p \geq 0.05$, two-sided t test). **f, g** Comparison of sequence spans and sequence timings between ramp conditions. **f** Determination of time span and center of sequences. To parameterize sequences, a line (red, also shown as dotted lines in **c** and **d**) is fitted to the sorted latencies (ordinate) across channels (abscissa). Here black dots denote the position of peak response latencies. Sequence center time is derived from the latency of the central point of the fitted line. Sequence span is determined by the latency difference between the two end points of the fitted line. **g** Average sequence spans (left, $n = 12$ conditions per ramp) and center time (right, $n = 12$ conditions per ramp) for different ramp conditions. Error bars denote 95% confidence level. Asterisks denote $p < 0.05$ in t test. Source data are provided as a Source data file.

nodes exhibits a remarkable inter-trial consistency and is independent of response kinetics. Thus, the rank order of response sequences could in principle be used to encode stimulus identity.

To examine how the temporal structure of the sequences scales with the duration of the ramps, we compared the span of the sequences (Fig. 1f) between ramp conditions. As expected, the sequence span increased with ramp duration and these differences were highly significant (no ramp: $122.59 \pm 10.86$ ms (s.e.m.), fast ramp: $201.33 \pm 8.70$ ms, slow ramp: $314.82 \pm 28.54$ ms. No ramp vs fast ramp: $t_{11} = -5.68$, $p = 0.00014$; no ramp vs slow ramp: $t_{11} = -6.70$, $p = 0.000034$; fast ramp vs slow ramp: $t_{11} = -4.68$, $p = 0.00067$; paired $t$-test. Figure 1g). However, this comparison revealed that the duration of the sequences does not scale proportionally with ramp duration. The duration of the slow ramp was twice as long as that of the fast ramp but the duration of the sequence increased only by a factor of $1.56 \pm 0.11$, which was significantly less than doubling ($p = 0.00068$, Wilcoxon signed-rank test). Thus, the time course of sequences was not a direct reflection of the increase in stimulus energy. The same was true for the dependence of peak latencies on ramp duration. As expected, increasing the duration of the ramp, i.e., slowing down the rise in intensity, delayed the onset of the sequences (Fig. 1g. no ramp: $155.69 \pm 1.92$ ms; fast ramp: $240 \pm 12.24$ ms; slow ramp: $319.25 \pm 27.24$ ms. No ramp vs fast ramp: $t_{11} = -7.39$, $p = 0.000014$; no ramp vs slow ramp: $t_{11} = -6.19$, $p = 0.000068$; fast ramp vs slow ramp: $t_{11} = -5.11$, $p = 0.00034$), but the changes were not proportional to the time course of the changes in stimulus energy. Latencies for the slow ramp increased only by a factor of $1.3 \pm 0.05$ rather than a factor of 2 when compared to the fast ramp ($p = 0.00049$, Wilcoxon signed-rank test). This disproportional scaling of sequence span and latency suggests that the two variables are not solely determined by stimulus parameters but depend also on network interactions. Further control analyses (Supplementary Notes, Supplementary Figs. 7 and 8) and simulation experiments (see below) support the notion that the sequences do not simply reflect the temporal structure of afferent signals or different sensitivities of the nodes to stimulus energy but are also a consequence of the dynamics emerging in cortical networks.

### Sequential neuronal responses enable fast identification of stimulus-specific information

After having characterized the stimulus specificity and the invariance of the sequences with respect to stimulus contrast and kinetics, we next investigated how stimulus-specific information is distributed over the duration of the sequences, or in other words,

how many nodes need to be activated to collect sufficient evidence for stimulus classification. To this end, we first verified that Bayesian decoders trained on response onset latencies (Supplementary Fig. 4c) could successfully predict stimulus identity (Fig. 2a and Supplementary Fig. 9a). This analysis confirmed the stimulus specificity of the response sequences. It should be emphasized that even the no ramp condition led to decodable response sequences. This implies that there was sufficient information about the sequence order of responses in the brief transient responses to permit decoding of stimulus identity. We then determined how early, on average, each channel (node) started to respond, and systematically included more and more of the fastest responding channels ("Methods"). The decoding accuracy exceeded chance level as soon as more than 4 to 5 channels were included. For the sudden onset stimulus, this required sampling over only 50 to 70 ms when the rank orders were derived from onset latencies (Fig. 2a and Supplementary Fig. 9a). As more and more channels were recruited, the decoding accuracy continued to increase. In comparison, when we used the firing rate vectors rather than the onset latencies of equal numbers of channels to decode stimulus identity, the decoding accuracy was much lower and required sampling over more than 180 ms to exceed chance level (Fig. 2b and Supplementary Fig. 9b). This comparison suggests that stimulus identity can be read out from onset latencies faster than from firing rates in the initial response phase.

We then used response peak times instead of onset latencies to test decodability from sequence order (Supplementary Figs. 10 and 11). Decoding accuracy based on peak times still required fewer channels to exceed chance level than that based on firing rates, especially in the high-intensity conditions (Supplementary Figs. 10 and 11). However, because peak latency is longer than onset latency by definition (minimum peak latency ~150 ms), more time elapsed until the minimal number of nodes were activated that allowed for above-chance classification. This reduced the advantage that decoding sequence order has over decoding rate vectors, as longer time enabled better estimation of firing rates. Indeed, when more channels were included and more time elapsed, decoding stimulus identity based on firing rates saturated at higher levels than decoding based on sequence order. In sum, these results indicate that stimulus identification from sequence order of responses can be achieved within very short intervals, even if only a small number of nodes are sampled; identification accuracy can be further improved by including information conveyed by firing rates if rate estimation is performed over longer integration intervals.

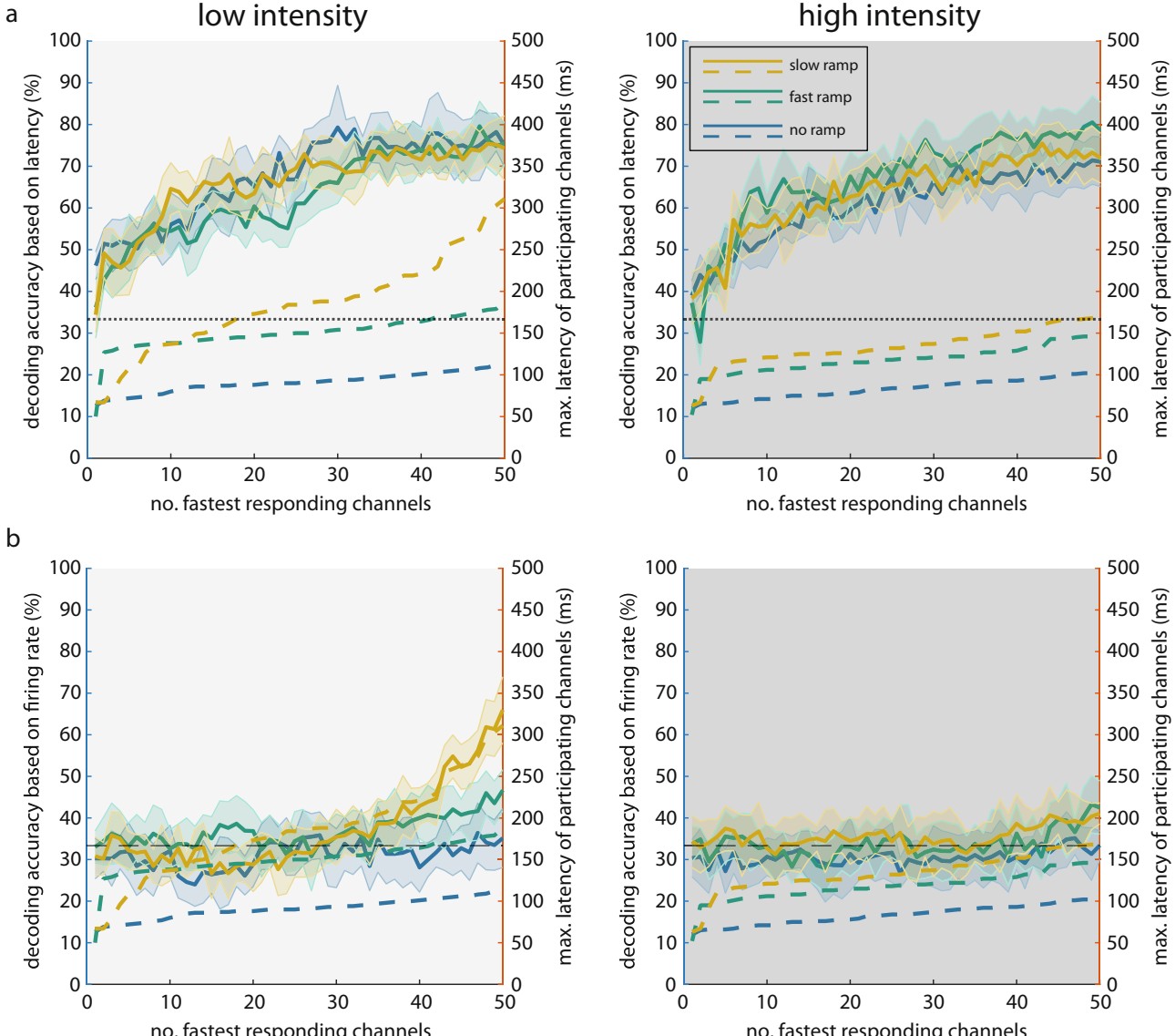

**Fig. 2 | Rank order of response sequences enables fast decoding of stimulus identity. a** Accuracy of decoding stimulus identity based on sequence order of onset latencies for low (left panel) and high (right panel) intensity conditions. Left ordinate (for solid curves): decoding accuracy based on sequences consisting of an increasing number of the fastest responding channels (abscissa). Shaded areas indicate 95% confidence interval around the mean. Horizontal black dashed line indicates chance level. Right ordinate (for dashed curves): average response latencies of the participating channels. **b** Similar to a but for the decoding analysis based on firing rates. Source data are provided as a Source data file.

## stimuli produce most informative response sequences

The observation that the temporal span of the stimulus-specific response sequences was considerably shorter than expected from the temporal evolution of the stimuli suggests that network interactions contribute to the generation of sequences. Afferent and intrinsic connections of the cortical networks are adapted through evolution and postnatal experience to capture characteristic features of the visual environment. Hence, priors required for the disambiguation of sensory evidence are stored in the functional architecture of the visual cortex (see "Introduction"). Therefore, we expected that the sequences might reflect not only the structure of the stimuli but also the extent to which sensory evidence matched the priors stored in the architecture of the cortical networks. To explore this possibility we created from the original natural images two further categories of manipulated stimuli: morphed images and phase-scrambled images (Fig. 3a). The morphed images were generated by applying a diffeomorphic transformation on the original three natural scene images[53], such that higher-order regularities like object identity and/or semantic information were removed, whereas some of the low-level features or elementary Gestalt principles such as continuity, collinearity and closure of contours were preserved. Scrambled images were created by adding random noise to the phases of the Fourier-transformed natural images, such that both high- and low-level features were eliminated. In the following we refer to stimuli as "natural" that conform to the statistical regularities of natural environments, irrespective of whether they represent novel or familiar objects. Thus, the three image categories, i.e., natural, morphed, and scrambled, differed by the degree to which the statistics of natural environments were preserved. Because our animals were raised in a normal visual environment, the stimuli were expected to match in a graded way the statistical priors stored in the connection architecture of the visual cortex.

To investigate the extent to which the match between sensory evidence and priors influenced the response sequences, we performed the same decoding analysis as above for the three stimulus categories ("Methods"). For all stimulus categories, the decoding accuracy was significantly ($p < 0.05$) above chance level (Fig. 3b, Supplementary

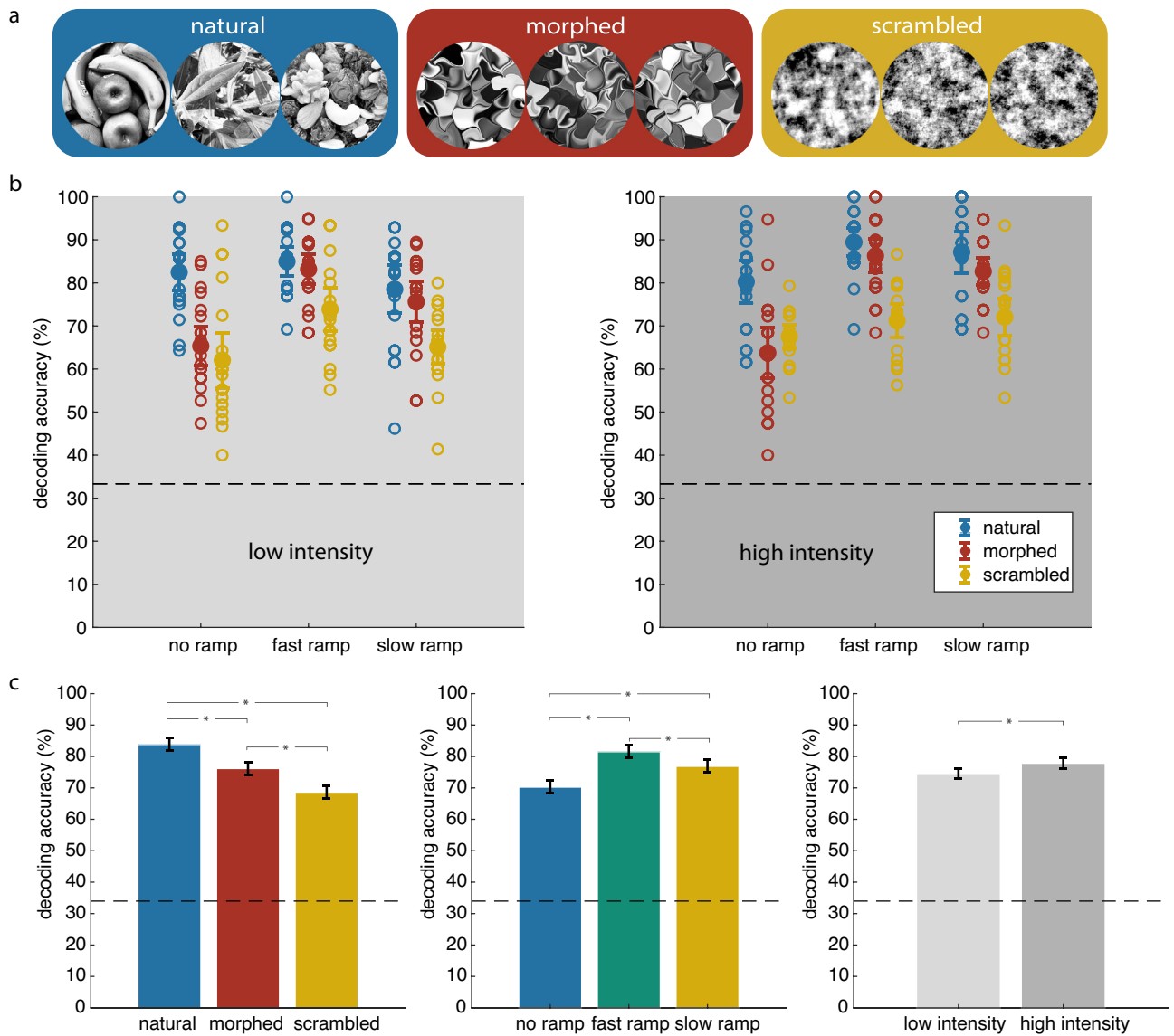

**Fig. 3 | Response sequences evoked by natural stimuli are most informative about stimulus identity. a** Stimulus categories. **b** Accuracy of decoding stimulus identity from sequences derived from peak response latencies. Left and right panels show the results for low and high-intensity conditions, respectively. Horizontal dashed lines mark chance level. **c** Average decoding accuracy marginalized over stimulus structure (left), ramp (middle), and intensity (right) conditions. Asterisks denote statistical significance ($p < 0.05$, two-sided t-test) of pair-wise comparisons. All error bars denote 95% confidence interval. Source data are provided as a Source data file.

Fig. 12), confirming the stimulus specificity of the sequence order of responses. However, the decoding accuracy differed between the stimulus categories (Fig. 3c. $F_{2,354} = 62.13$, $p < 0.01$, ANOVA). Accuracy was highest for natural images (83.80 ± 0.96%, s.e.m.), lowest for scrambled images (68.63 ± 0.96%) and intermediate for morphed images (76.14 ± 0.96%). All pair-wise comparisons were Bonferroni-corrected for multiple comparisons and significant at the $p < 0.05$ level. We replicated the decoding analysis using response onset rather than peak latencies for the training of decoders and obtained qualitatively identical results (Supplementary Fig. 13). Thus, although the sequence order of response latencies was specific for individual stimuli in each category, the degree of specificity depended on the statistical structure of the stimuli. The better the stimuli matched natural image statistics, the better was decoding accuracy. Furthermore, the decoding accuracy was higher for the ramp than the no ramp conditions (Fig. 3c. No ramp: 70.23 ± 0.96%; fast ramp: 81.49 ± 0.96%; slow ramp: 76.85 ± 0.96%. ANOVA $F_{2,354} = 34.64$, $p < 0.01$). All post hoc pair-wise comparisons were significant ($p < 0.05$). Thus, the better

segregation of onset and peak latencies caused by the ramping procedure facilitated decoding. In this context it is noteworthy that the decrease in decodability in the no ramp condition was minor for natural stimuli and somewhat more pronounced for the manipulated stimuli, in particular for the scrambled stimuli. A likely interpretation is that sequences evoked by unnatural stimuli are less precisely timed than those induced by natural stimuli and therefore lead to more ambiguities when compressed.

Because these results were obtained in a higher visual area (V4), we considered it important to examine whether they also held true for V1 and therefore repeated the analyses on a dataset obtained from V1 in two other monkeys. The results were similar (Supplementary Fig. 14).

**Recurrent network model with Hebbian synapses recapitulates prior-dependent sequence refinement and stabilization**
Recurrent networks respond to perturbations with sequential activation of their nodes but in the presence of background activity

consistent and robust reproduction of particular sequences is not trivial. The network connections must be anisotropic in order to permit propagation of activity along a constant path. Such anisotropy can be generated either by design or by learning mechanisms that change the gain of the recurrent connections. To explore the possibility that learning stabilizes temporal sequences in a stimulus-specific way, we

trained a spiking neural network in a non-supervised way to acquire information about the shape of a digit (Fig. 4. "Methods"). The synapses of the recurrent connections were made susceptible to activity-dependent modifications by endowing them with spike timing-dependent plasticity (STDP)[54–56]. Thus, repeated stimulation is expected to strengthen and weaken connections between neurons

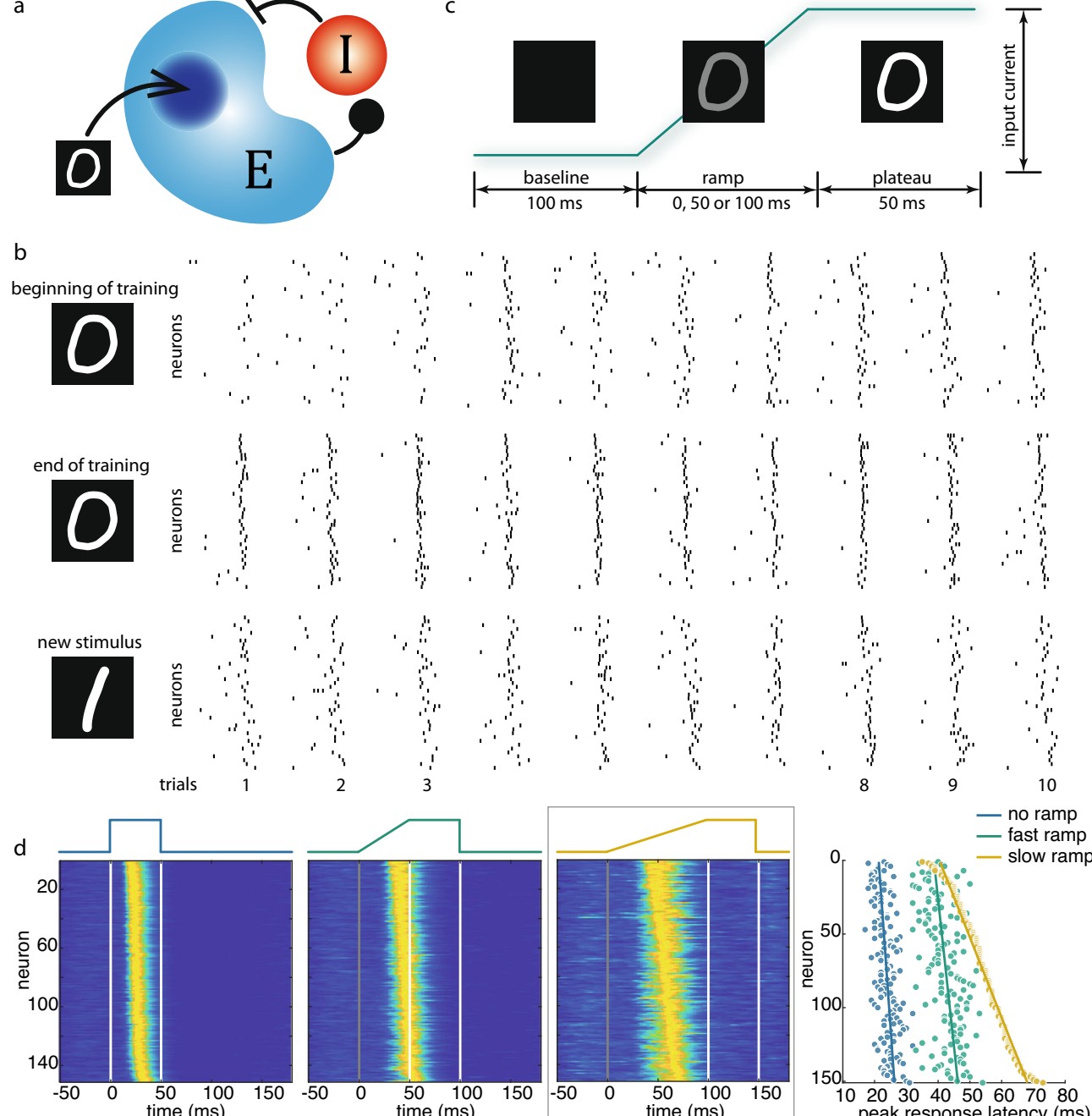

**Fig. 4 | Spiking neural network model replicates stimulus-evoked response sequences. a** Schematic of the model. The network consists of excitatory (blue, $N = 200$) and inhibitory (red, $N = 50$) neuron groups that are interconnected both between and within groups. External input is mapped to a subset of excitatory neurons (dark blue, $N = 49$). **b** The effect of exposure on the precision of sequences. The raster plots show multiple single-trial population spiking responses at the beginning (upper row) and end of the training (middle row, 200 training trials in total), evoked by the same training stimulus (digit "0"). The lower row shows the population responses of the trained network evoked by a new stimulus (digit "1"). We only evaluated those excitatory neurons that did not receive external input

directly ($N = 151$). For clarity, only 40 of them are shown. **c** Simulation of ramping conditions. The image pixel intensities are mapped to the input current of individual neurons. **d** Network responses to ramping stimuli. With ramping stimuli, response transients unfold into sequences, as in the experiment. The responses are sorted based on the latencies of the peak responses in the slow ramp condition (enclosed by gray box). Correlations between peak response latencies and neuron indices (as quantification of the steepness of the lines fitted to the response latencies) were significantly ($p < 0.05$) greater than zero for all three ramp conditions (no ramp: $r = 0.49$, $p < 0.01$; fast ramp: $r = 0.43$, $p < 0.01$; slow ramp: $r = 0.99$, $p < 0.01$). Source data are provided as a Source data file.

depending on the temporal order of their activation. For training we used only sudden onset stimulus. Already at the beginning of the training, the networks produced sequences but these were highly variable. With training the sequences evoked by a particular stimulus became more robust, precise, and reproducible. For further comparison with the experimental results we examined whether sequence order was maintained when the sequences were expanded in time by presenting ramping stimuli. We simulated the ramping stimuli as linearly increasing input current to the neurons. As in the experiments, the span of the sequences increased without changes in sequence order. When networks trained on a particular stimulus were presented with a different, novel stimulus, the sequences evoked by the new stimulus were more spread out in time and showed more inter-trial variability than the sequences evoked by the "familiar" stimulus. This simple and by no means comprehensive proof-of-concept simulation suggests the possibility that the robust temporal sequences observed in the experiments resulted from interactions in a recurrent network in which the strengths of the coupling connections had been shaped by experience (see above).

### Natural stimuli enhance persistence of stimulus information

So far we have focused exclusively on the sequence order of responses in the initial phase of evoked activity, but this initial transient is followed by sustained, slowly decaying activity. This persistent activity is to a large extent due to continuing sensory drive but likely reflects also activity contributed by intracortical interactions. It is to be expected, therefore, that the sustained activity, and the information it carries, is also influenced by the degree to which the stimulus matches the priors stored in the synaptic weights of the cortical network. To test this possibility we trained independent decoders at successive time points (sliding window width 100 ms, step size 100 ms) to predict stimulus identity from the firing rate vectors of the population responses ("Methods"). This analysis was performed again separately for natural, morphed, and scrambled stimuli.

For all stimulus conditions (different onset time courses and stimulus categories), decoding performance was maximal for the response transient (Fig. 5a, b, Supplementary Fig. 15), confirming that the bulk of stimulus-specific information can be retrieved from a short stretch of the initial response phase. Notably, in the ramp conditions, the decoding performance peaked well before the ramp reached its maximal intensity, suggesting that sufficient sensory information had already accumulated to produce a ceiling effect before the stimuli reached full contrast. In this transient response phase decodability of the three stimulus categories was similarly high and showed no significant difference between the three stimulus categories. Stimulus-specific information reached its maximum at the same time as the firing rate (average per channel, Fig. 5a, b). This was the case even though stimulus intensity kept increasing beyond the peak of firing rates in the ramp conditions. During the sustained response phase decodability of stimulus identity decreased with elapsing time from stimulus onset for all stimulus conditions. However, now differences became apparent between natural and manipulated stimuli. The decay of decoding accuracy was slowest for natural images, fastest for scrambled images, and intermediate for morphed images, leading to significantly different decoding accuracies for the three stimulus categories (Fig. 5a, b, Supplementary Fig. 15). To quantify the time course of the fading of stimulus-specific information, we fitted a linear function to the descending phase of each decoding accuracy curve and took its slope as indicator of the decay speed of stimulus information (insets in Fig. 5a, b). This decay was significantly influenced by the stimulus structure ($F_{2,354} = 38.99$, $p < 0.01$, ANOVA) and ramp condition ($F_{2,354} = 112.86$, $p < 0.01$) but not by stimulus intensity ($F_{1,354} = 2.17$, $p = 0.14$). The decay of stimulus-specific information was slowest for natural images ($-34.69 \pm 2.66\%$/s, s.e.m.), fastest for scrambled images ($-66.96 \pm 2.66\%$/s), and intermediate for morphed images

($-43.76 \pm 2.66\%$/s). All post hoc pair-wise comparisons were statistically significant at the $p < 0.05$ level (Fig. 5c). These differences in decodability were not simply due to differences in sustained firing rates, because firing rates were consistently highest for morphed images across all ramp and intensity conditions (lower panels in Fig. 5a, b). This implies that there is no straightforward correlation between information content and the firing rate of the population response. Both findings, the prolonged persistence of information about natural stimuli and the lack of correlations between decodability and average firing rate, have been reproduced in area V1 (Supplementary Fig. 16).

### Temporal evolution of population responses

In order to obtain an intuition for the decoding results based on rate vectors, we projected the high-dimensional vector of population activity into the low-dimensional space spanned by the three largest principal components (Supplementary Fig. 17) and plotted the trajectories that represent the temporal evolution of the population activity (Fig. 6 and Supplementary Fig. 18. "Methods"). To calculate firing rates with high temporal resolution, we used short sliding windows of 30 ms duration that were moved in steps of 1 ms. The plots represent trajectories averaged across trials. After stimulus onset, the trajectories of population activity diverted rapidly from a common baseline region, followed stimulus-specific paths (Fig. 6a) and settled at different endpoint regions. Measurements of the Euclidian distances between population activity vectors evoked by different stimuli and the distances of the respective vectors from baseline revealed that the trajectories for the natural stimuli remained well separated from each other throughout the whole stimulation period (Fig. 6a). By contrast, for scrambled images the trajectories were less separated and returned more rapidly to the regions close to baseline.

Stimulus discriminability likely depends not only on the distances between trial-averaged trajectories but also on their trial-to-trial variability. Therefore, we calculated a discriminability index ("Methods") by taking the ratio between inter-trajectory distance and trajectory variability at each time point to assess the stimulus specificity of the trajectories. This measure reproduced the results of the decoding analysis (c.f. Fig. 5): the discriminability index was highest for natural, lowest for scrambled, and intermediate for morphed stimuli (Fig. 6b). The discriminability index started to rise within 100 ms to 200 ms after stimulus onset and reached a maximum within 200 to 400 ms, depending on stimulus intensity and ramp duration (Fig. 6b). These intervals reflect the fast segregation of the response trajectories (Fig. 6a) and they also correspond to the duration of the response sequences (c.f. Fig. 1f). In agreement with the time course of decodability from sequence order (c.f. Fig. 2) this suggests that stimulus-specific information is readily available and decodable long before the trajectories of the rate vectors reach a stable endpoint. Comparison of the rising phases of the discriminability indices showed that discriminability increased consistently faster for the natural than the manipulated stimuli (Fig. 6b). This suggests that the priors stored in the functional architecture of cortical circuits contribute already to the shaping of the very early transient response components; and this is in agreement with the finding that natural stimuli evoked response sequences whose decodability was better than that of sequences evoked by manipulated stimuli. Taken together, both the decoding results from sequence order and from rate vectors support the conclusion that decodable stimulus-specific information results from an interaction between sensory evidence and stored priors and reaches a maximum already during an early phase of the transient response.

## Discussion

The present results indicate that the peak and onset latencies of the early transient responses of cortical neurons are organized in sequences whose order is stimulus-specific and preserved even when the absolute latencies of the responses and the duration of the

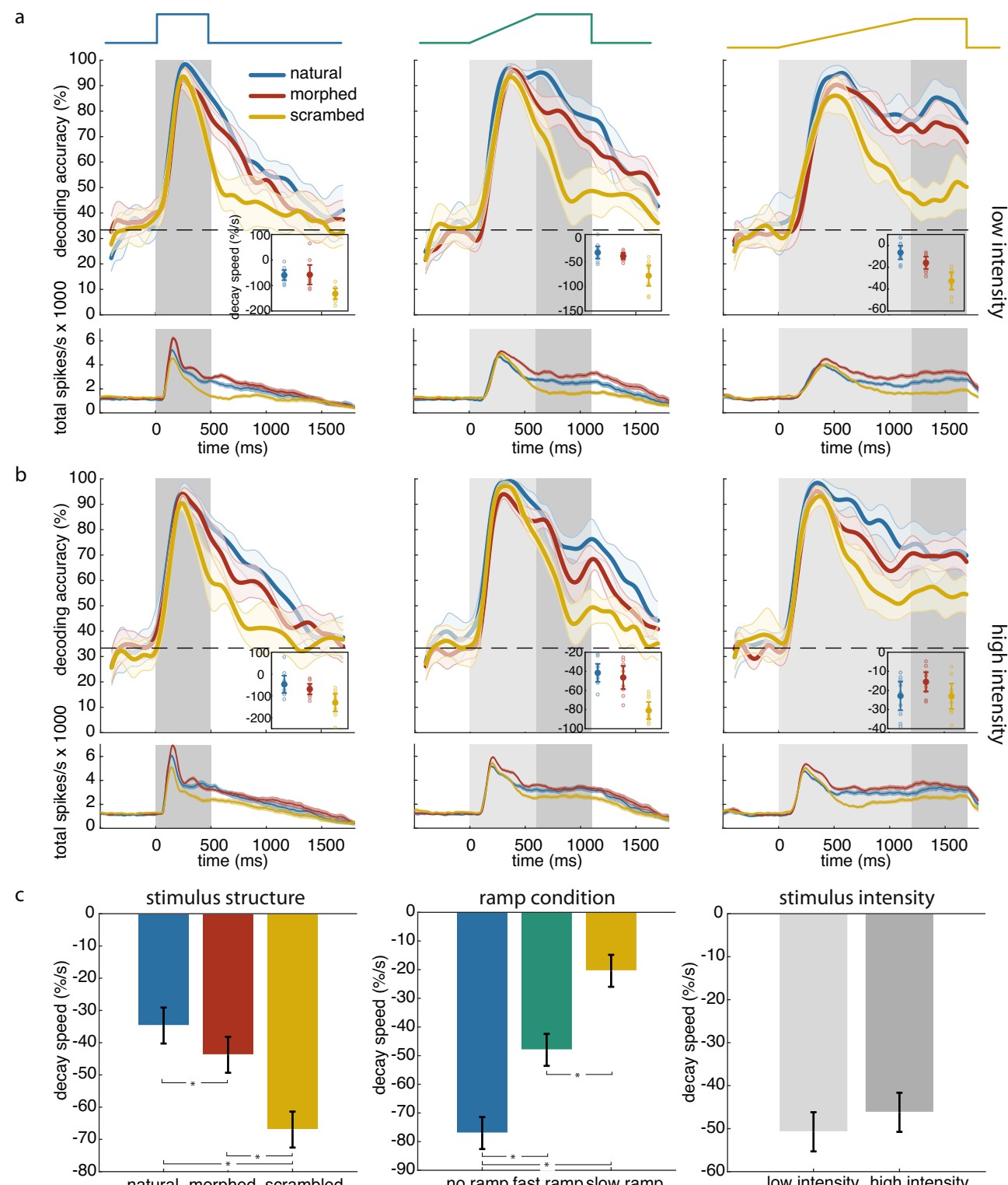

**Fig. 5 | Natural stimuli extend the persistence of stimulus-specific information.**
**a** Effect of stimulus structure on the time course of stimulus-specific information
and firing rate. Upper panels: time-resolved decoding accuracy of stimulus identity
based on firing rate vectors. Inset: decay speed of decoding accuracy. Lower
panels: corresponding average firing rates per channel. The three columns refer to
no ramp (left), fast ramp (middle) and slow ramp (right) conditions. Horizontal
dashed lines indicate chance level. Shaded areas around the curves denote 95%
confidence interval around the mean. Light and dark gray areas mark the ramp and
plateau periods, respectively. **b** Same conventions as in (**a**), but for the high-
intensity conditions. **c** Decay speed of stimulus information as function of stimulus
structure (left), ramp condition (middle) and stimulus intensity (right). Asterisks
mark statistical significance ($p < 0.05$, two-sided t-test) of pair-wise comparisons.
Error bars indicate 95% confidence level around mean. Source data are provided as
a Source data file.

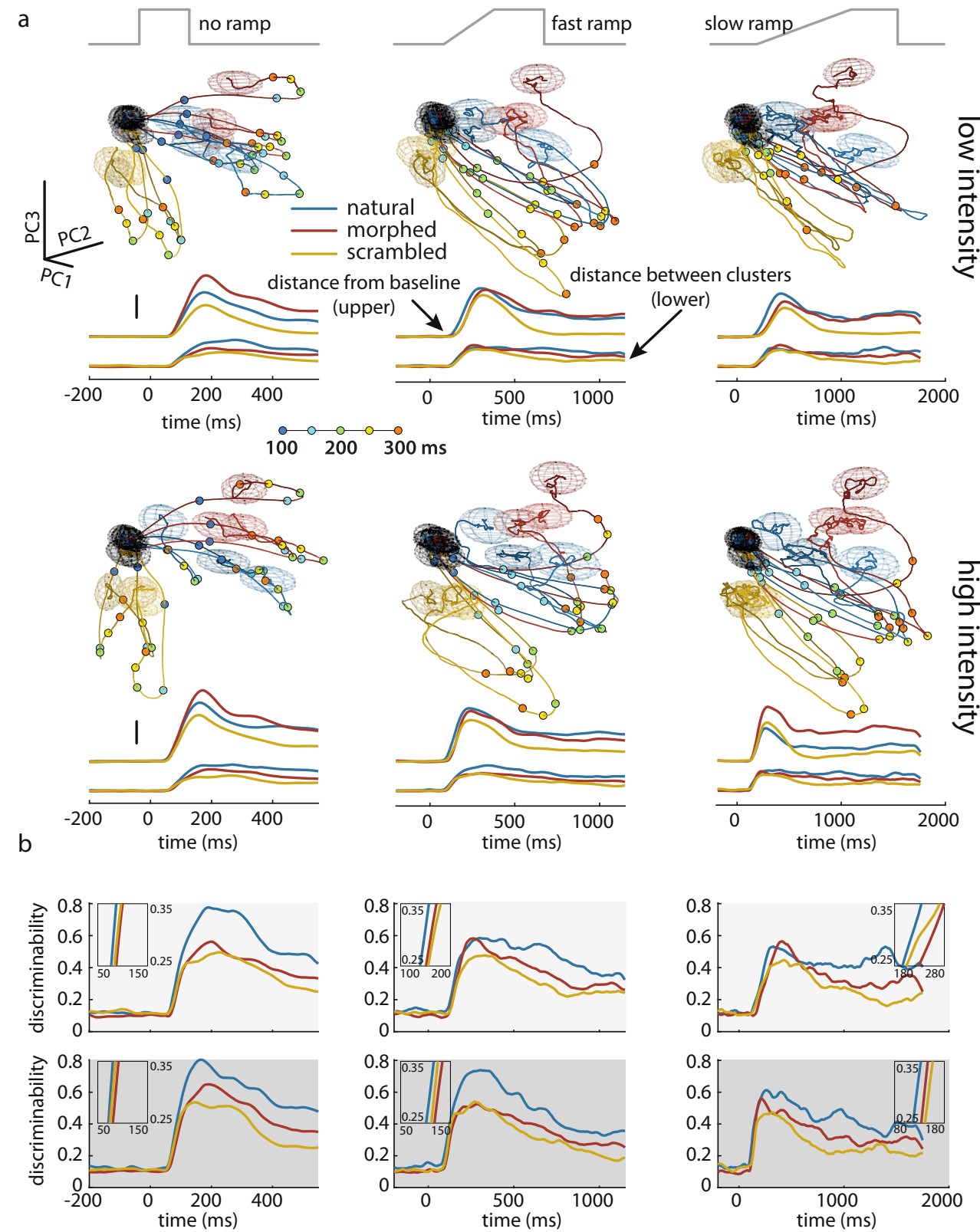

sequences are changed by manipulating the intensity and the time course of the stimuli. The duration of the sequences increased with ramp duration and decreasing stimulus intensity but sequences remained compact. Their dilation did not scale proportionally with ramp duration and the response time of neurons in the sequences was neither locked to stimulus intensity nor to the temporal integral of input drive. This suggests that the sequences are not solely due to

differences in the feature selectivity and contrast sensitivity of the recorded neurons. The finding that decodability of stimulus-specific information from sequence order rose faster and reached higher levels for natural than for morphed and scrambled stimuli suggests that the time course and rank order of the sequences were not solely determined by the physical structure of the stimuli but in addition reflect the degree to which sensory evidence matches priors about the

**Fig. 6 | Visualization of the time course of stimulus-specific trajectories of neural population activity. a** 3D representation of population firing rate vectors in principal component space (calibration scales in the first panel: length = 100 in each PC axis, dimension = "spikes/sec"). Trajectories (one trace per stimulus, averaged across trials) are separated into panels referring to different ramp (columns) and intensity (rows) conditions. The colors of the trajectories refer to the three stimulus categories. Oval meshes show the covariance (scaled for visualization) of the baseline (black, sampled over 200 ms before stimulus onset) and of the end points (in the respective colors, sampled over the last 200 ms of the plateau period) of the stimulus-specific trajectories. Color dots on the trajectories mark the time points

between 100 and 300 ms in steps of 50 ms. Traces below the 3D plots show the time course of the distance between the trajectories and the baseline (upper) and the separation between different trajectories (lower), both calculated separately for each stimulus category and represented in the respective colors. Vertical calibration bars mark the length of 200 in the dimension of "spikes/sec". **b** Discriminability index for the trajectories in each stimulus category. Arrangement of the six panels is the same as in (**a**) (columns for ramp conditions, rows for intensity conditions). Insets zoom onto a segment (range indicated on the frames) of the rising phase of the discriminability curves.

statistical properties of natural scenes. This suggests that the shaping of the sequences that increased their decodability occurred at the cortical rather than the subcortical level of the processing stream. Several arguments suggest recurrent interactions as a likely substrate for this matching process: (i) the network of recurrent, "patchy" connections in the visual cortex is shaped by experience and hence contains priors about the statistics of natural images; (ii) recurrent networks respond to perturbations with a sequential activation of their nodes; and (iii) our simulations of a simple recurrent network with spiking neurons and recurrent connections endowed with Hebbian synapses reproduced the beneficial effect of experience on the decodability of sequences. As expected, the untrained, "naive" network already generated sequences upon stimulation but the precision and reproducibility of the sequences increased when the network was familiarized with the stimulus through non-supervised learning. Moreover, the rank order of the sequences was preserved when they were expanded or compressed in time. The hypothesis that responses were shaped by intracortical interactions that are most likely mediated by recurrent networks is further supported by the finding that the persistence of stimulus-specific information in the firing rate vectors was longer for responses to natural stimuli than their manipulated counterparts. The underlying assumption is that stimulus-specific dynamic states resonate better and reverberate for longer intervals if induced by stimuli whose statistical regularities match stored priors rather than if induced by stimuli that correspond less to the statistical regularities of natural scenes (see below). Taken together these observations suggest that the sequences were generated by recurrent interactions at the cortical level. However, as detailed in the next paragraph, we cannot exclude other mechanism.

The response latencies were assessed from stimulus locked increases in firing rates and therefore may not reflect the exact timing of the very first evoked spikes. However, similar methods are commonly used to estimate response latencies in electrophysiological[41,57,58] and optical recordings[59,60]. We derived response latencies both from measurements of response peak latencies and from threshold crossing time of firing rates. Both methods gave very similar and highly consistent results. Thus, if the rank order of sequences were used as a code for stimulus identity, putative readouts could rely on both variables.

Another critical point is the interpretation of the decoding results. The implicit assumption is that the stimulus-specific information retrieved by the decoders from sequence order and rate vectors does actually support perception. The current experiments were performed under passive viewing conditions and the monkeys were rewarded to keep fixation rather than identifying the stimuli. Thus, we had no possibility to establish correlations between perceptual decisions and the performance of the decoders. The finding that decoders performed less well on manipulated than natural stimuli suggests, however, that they extracted behaviorally relevant information. Evidence from psychophysical studies indicates that natural stimuli are recognized faster and better than their manipulated counterparts[61–64].

Our results do not allow us to unequivocally determine the mechanism responsible for the generation of sequences and to determine the extent to which they were generated by cascades of feedforward processing and by recurrent interactions, respectively. To

resolve this question, experiments will have to be performed on a larger sample of natural and manipulated stimuli and responses need to be obtained along the whole processing stream, including retinal, geniculate and cortical responses, the latter being assessed with laminar resolution. It would then be possible to identify at which processing level sequences emerge that comply with our results; i.e., sequences that (i) remain invariant to temporal compression and dilations caused by stimulus kinetics and contrast, (ii) are insensitive to the small eye movements which occur even under fixation conditions and (iii) are sensitive to the statistics of natural scenes.

Visual perception is fast. Humans can parse cluttered visual scenes and recognize objects within about 150 ms[46]. If this process involves sequential computations across a hierarchy of brain areas in the visual pathway, as is commonly assumed[65,66], it leaves only tens of milliseconds for computation at each processing stage[46]. Given the sparse activity of cortical pyramidal cells, individual neurons can only contribute a few spikes in these short intervals. As pointed out by several authors[32–34,42,67–71], this raises a problem if information were solely encoded in the discharge rate of neurons. Spikes would have to be summed over long intervals to distinguish differences in discharge rate which is in conflict with processing speed. Alternatively, discharges of a large number of neurons would have to be evaluated jointly to perform a maximum likelihood estimation[72]. However, this strategy is hampered by the fact that the discharge rate variabilities of neurons are not independent but exhibit correlations, so called "noise correlation"[73]. Thus, averaging would also increase the noise component.

An alternative strategy is to encode information in the relative timing of responses because such codes can be read out rapidly by neurons that are either sensitive to coincident[74–77] or sequential input[78–82]. Evidence that the timing of discharges matters in neuronal processing is available from studies in various systems: vision[57,67,83–88], audition[74,76], olfaction[41,58–60,89–92], somatosensation[93–96], spatial navigation[97–100], and motor control[101–105]. According to our results, decoders capable of evaluating sequence order can correctly classify natural stimuli with sudden onset within 70 ms after stimulus onset. Even for the low-intensity slow-ramp stimuli, stimulus-specific information emerged within 100 ms and plateaued within 200 ms. Thus, the time required for correct identification was extremely short and less than the duration of the whole sequences. A decoding mechanism based on response timing can therefore trade between processing speed and accuracy by successively sampling over sequentially activated nodes. As our data suggest, decoding of response timing is more efficient than decoding of rate vectors both with respect to speed and required number of nodes. The trajectories derived from rate vectors induced by different stimuli began to diverge only at times when sequence order could already be decoded from the fastest responding nodes; and maximal separation of trajectories occurred only at times at which most neurons in a sequence have already responded.

Our data are not only in agreement with a temporal code in form of staggered response latencies, but also demonstrate in addition that stimulus identity can be decoded from the *rank order* of sequentially activated nodes that is independent of absolute timing. Using rank order as template we could decode stimulus identity irrespective of

the overall time course of the sequences and the intervals between the nodes' activation. If rank order rather than staggering of absolute response latencies were used as code for stimulus identity, this would allow readout without requiring an independent reference signal that defines stimulus onset[33]. Such a reference signal could be provided during active vision by the corollary activity associated with saccadic eye movements because the arrival of new sensory evidence is predicted by the time of fixation onset. Evidence indicates that this corollary activity reaches primary visual cortex and is precisely time-locked with the saccades[106–108]. However, there are many instances when the time of arrival of new sensory evidence cannot be predicted, rendering readout of codes difficult that rely only on stimulus locked absolute latencies. This problem could be overcome if the readout layer consisted again of a recurrent network as such networks are ideally suited to decode and process sequential input signals and/or time series[17].

Coding information in the rank order of sequentially activated nodes is also an economical strategy because the code is permutable. As our data indicate, the very same nodes can participate in different stimulus-specific sequences which economizes the number of nodes required for the representation of different input patterns. Moreover, it exploits not only space, the spatial distribution of activated nodes, but also time, the sequence order of activated nodes, as coding dimension which further expands coding space.

Recurrent networks sustain reverberating activity and traveling waves[18,20,22] and therefore respond to stimuli with sequential activation of their nodes. However, further analysis of recurrent dynamics is required to identify the mechanism that can generate sequences at widely different timescales while preserving the rank order of sequentially activated nodes. This question is of general interest since similar observations have been made in other brain areas. During ripple oscillations, firing sequences of hippocampal place cells, generated by previous navigation at behavioral timescales, are replayed in the same or reversed order but much compressed in time[99,109,110]. Similar observations have been made in the medial frontal cortex; the temporal profile of population responses can be stretched or compressed to span different time intervals[111].

Networks operate at variable timescales[112,113], and these timescales are determined by a number of extrinsic and intrinsic variables. Some of the intrinsic variables are unlikely to change at the short timescales required to account for the expansion and contraction of sequences. These comprise the connectivity[114] and cell-intrinsic properties[115] such as synaptic kinetics[116] and ion channel and receptor compositions[117–120]. However, the highly non-linear interactions in recurrent networks enable changes in their dynamics to occur at short timescales when perturbed by external input[121] or when the E/I balance is modified[122,123]. Furthermore, in recurrent networks whose nodes have a propensity to oscillate, additional mechanisms can determine the pace of sequential activation. In delay-coupled oscillator networks, resonance and interference phenomena, phase shifts, and traveling waves as well as changes in oscillation frequency are in principle capable of maintaining fixed orders in the sequential activation of the units at variable timescales[124–128]. The highly non-linear dynamics of delay-coupled oscillator networks may thus actually contribute to the stabilization of sequences despite compression or expansion of their overall duration. How the information contained in sequence order can be encoded and decoded despite variations in the sequences' duration is a fundamental problem, e.g., in the production and perception of speech[129,130], and requires further investigation. We suggest that recurrent networks in down-stream cortical areas should be able to extract the information conveyed by rank order codes once they are trained to detect their invariant properties.

The accuracy of decoding stimulus identity from sequence order was highest for natural images, suggesting that the sequences produced by natural stimuli are more informative than those induced by morphed or phase-scrambled stimuli. Moreover, the decay of stimulus-specific information in the rate vectors of the sustained response phase was slower for the natural than the manipulated stimuli. We propose that the better decodability of responses evoked by natural stimuli is the result of a good match between sensory evidence and the priors stored in the functional architecture of the cortical network. As detailed above, the reciprocal connections between columns responding to features which have a high probability of being correlated in natural environments get strengthened through a Hebbian mechanism[6,24,27,31,131,132]. Thus, the coupling strengths of the recurrent connections constitute a physical embodiment of natural scene statistics and could serve as priors in perceptual inference[17]. These priors are likely to stabilize reverberating responses evoked by stimuli whose statistics match the stored model of the visual world. The reason is that such stimuli drive nodes or columns that are strongly coupled and hence can sustain reverberating activity. Our simulations seem to support this notion. Because of "experience"-dependent Hebbian modifications the recurrent network connections adapted to the "statistics" of the repeatedly presented stimulus. The consequence was, that this stimulus evoked more precise and consistent sequences than new stimuli, to which the network had not been adapted previously. However, due to the small size of the network and the restricted set of training and test stimuli it was difficult to distinguish between "learning about the general statistics of stimuli" and "learning to represent a particular stimulus".

It has been proposed that sensory evidence that matches priors induces a rapid change in network dynamics towards low-dimensional sub-states[133] and it has been shown that these sub-states exhibit stimulus-specific correlation structures and reduced variability of discharge patterns[134]. Our analysis of response trajectories indicated a rapid divergence from the baseline. The time course of this divergence corresponded to the duration of the response sequences. Thus, the generation of the sequences could be a reflection of the network's descent from high-dimensional resting activity towards a lower dimensional and stimulus-specific dynamic sub-state. Such a fast and early interaction between sensory evidence and stored priors is further supported by the fact that decoding performance based on sequence order depended on stimulus structure already during the very early transient response phase; the rise time of stimulus discriminability was faster for natural stimuli than for stimuli likely to match less well with stored priors. Taken together, these considerations suggest that both the short response sequences and the subsequent sustained responses are shaped by a Bayesian matching operation that compares sensory evidence with the priors stored in the cortical networks, and this matching operation seems to set in right at the beginning of the responses.

As discussed above, encoding information in neuronal response sequences enables fast readout of stimulus-specific information even when only few nodes have been activated. By directly comparing decodability based on sequence order vs firing rate vectors in the transient response phase, we showed in addition that as time elapses decodability based on firing rates saturates at higher levels than that based on sequence order. Firing rate decoders benefit from longer integration time and allow for more accurate identification of stimulus-specific information. Thus, rate and temporal codes could complement one another, trading speed against accuracy.

The time-resolved decoding results also indicate that the stimulus-specific information contained in the rate vectors is maximal during the transient response component and decays during the sustained response phase. The finding that this information persisted over longer intervals for responses evoked by natural than degraded stimuli highlights another aspect of the dynamics reflecting the interaction between sensory evidence and priors. The persistence of stimulus-specific information in the sustained response component is of course to a large extent due to the continuous presence of the stimulus, but is

also influenced by the intrinsic dynamics of the recurrent network[12,135]. As shown by Berkes et al.[136] and Fiser et al.[137], the correlation structure of intrinsic cortical activity shares features with the activity patterns evoked by natural stimuli. The longer persistence of stimulus-specific information in the rate vectors evoked by natural rather than manipulated stimuli is thus likely due to a good match between sensory signals and intrinsic network dynamics. Interestingly, however, there was no direct correlation between stimulus information and the amplitude of the rate responses. In general, decodability of response vectors increases with discharge rate[135]. However, in the present case morphed stimuli produced the highest firing rates and their trajectories traveled the longest distances, but their decodability was lower than that of natural stimuli and decayed faster. For the phase-scrambled stimuli, by contrast, the relations between decodability, amplitude, and persistence of rate responses were as expected. They evoked the lowest firing rates and their trajectories returned rapidly towards the baseline. Thus, further analysis is required to determine which features of the decaying discharge rates contributed to the decodability of stimulus-specific information.

Taken together, the results of the present study suggest the possibility that the cerebral cortex complements the feedforward processing of rate-coded information by exploiting the dynamics of recurrent networks to generate a temporal code that represents stimulus-specific information in the rank order of sequentially activated nodes. The properties of this temporal code comply with the criteria for an efficient and sparse code. First, it compresses relevant information in short time intervals and therefore could allow for ultrafast readout. Second, it is permutable and allows the same neurons to participate in the encoding of different stimuli, which economizes hardware. Third, it is invariant and robust with respect to stimulus properties such as intensity, time course, and exact retinal position. The latter follows from the fact that even under conditions of visual fixation small eye movements (microsaccades) persist. Forth, it captures the match between sensory evidence and priors. Hence, information encoded in sequences could be exploited in parallel to rate modulations to convey complementary information. In order to ascertain that this temporal code is actually used by the brain, causal evidence has to be obtained for the behavioral relevance of this putative coding strategy. This is a challenge for future studies because it requires interference with the timing of the nodes' responses without at the same time altering their discharge rate and to correlate the effects of such manipulations with the animal's ability to distinguish between stimuli.

## Methods

### Electrophysiology

The experiments were conducted on four rhesus monkeys (*Macaca mulatta*): one female (Monkey I, 9 kg, 17 years old) and three males (Monkey K, 12 kg, 10 years old; Monkey H, 17 kg, 11 years old; Monkey A, 12 kg, 13 years old). All experimental procedures were in compliance with the German and European regulations for laboratory animal protection and welfare, and were approved by the local authority (Regierungspräsidium Darmstadt). Monkeys H and K were implanted in the left hemisphere with two CerePort Utah Arrays (Blackrock Microsystems, Salt Lake City, Utah, USA), one in area V1 and the other in area V4. Each array had 64 microelectrodes. Only the V4 electrodes were considered in the main experiment. Monkeys A and I were implanted over V1 with a Microdrive system (Gray Matter Research, Bozeman, Montana, USA) that contained 32 individually adjustable microelectrodes and provided the data for the measurements in area V1.

### Behavioral paradigm

The monkey was seated in a custom-made primate chair inside a dark experimental booth. The distance between the eyes and the stimulus monitor (Samsung SyncMaster 2233RZ; 120 Hz refresh rate) was 80 cm for monkeys H and K, and 60 cm for monkeys A and I. The eye position was monitored using the EyeLink tracker (SR Research, Ottawa, Ontario, Canada). During recording, the monkey performed a passive viewing fixation task. The monkey initiated a trial by fixating a white fixation point presented at the center of the screen. During the entire trial, the monkey had to maintain fixation within a window of about 1 degree of visual angle. During this fixation period a visual stimulus was presented on the screen. Successful maintenance of fixation throughout the trial was rewarded with a drop of juice or water at the end of the trial.

In the main experiment, the structure of a trial was as follows. After a minimal interval of 500 ms from the beginning of fixation (pre-stimulus baseline period), the stimulus was presented and remained on the screen until the end of the trial, when both the stimulus and the fixation point disappeared simultaneously. At stimulus onset, we ramped the stimulus intensity gradually such that the stimulus appeared slowly on the blank gray background until it reached its maximal intensity. This was achieved by linearly increasing the blending ratio (transparency or alpha value) between the stimulus and the gray background. We used three ramp conditions: slow ramp, fast ramp, and no ramp, the latter corresponding to sudden stimulus onset. The respective ramp durations were: 1000 ms and 500 ms for monkey K, and 1200 ms and 600 ms for monkey H. We used two different levels for maximal intensity corresponding to alpha values of 10% (low intensity) and 30% (high intensity), respectively. Once maximal intensity was reached, the stimulus remained at the respective level for at least 500 ms (plateau time). The durations of the pre-stimulus baseline and the plateau phases were semi-randomized to reduce predictability of the trial time course. The total trial length varied between 2200 ms to 3000 ms. The end of a trial was signaled by the disappearance of the fixation point.

In addition to the three ramp and two intensity conditions, three different categories of stimuli were presented, amounting to 18 conditions in total. All experimental paradigms were implemented using the MATLAB-based in-house software ARCADE[138]. In total, monkey H performed 2955 trials in 6 sessions/days ($492.5 \pm 113.3$ (s.d.) trials per session, or $159.8 \pm 25.1$ trials per condition); monkey K performed 4679 trials in 6 sessions ($779.8 \pm 209.9$ trials per session, or $253.3 \pm 48.4$ trials per condition).

In the V1 experiments, we simplified the experimental conditions, and used only one ramp duration (2000 ms) and one intensity level (50%), and kept only three conditions of stimulus category. Because monkeys I and A have been previously trained for a different task that required behavioral response with a mechanical lever, our experimental paradigm had to be adapted as follows. The appearance of the white fixation point signaled the beginning of the trial. After a baseline of 1000 ms, the stimulus was presented in linearly increasing intensity from 0 to 50% over the interval of 2000 ms. Then the fixation point changed color to either green or blue, and the monkey needed to respond to the color change by moving a two-way mechanical lever forward or backward, respectively. The monkey was required to maintain fixation until giving the correct lever response in order to get a reward. In total, monkey A performed 3133 trials in 3 sessions ($1044.3 \pm 52.7$ trials per session, or $1044.3 \pm 1.1$ trials per condition); monkey I performed 2418 trials in 3 sessions ($806.0 \pm 391.6$ trials per session, or $806.0 \pm 2.0$ trials per condition).

### Stimulus design

In the V4 experiments, the stimuli were presented on gray blank background, had a circular shape, were $500 \times 500$ pixels (10° visual angle) in size, and were positioned such that both the fixation point and the classical receptive fields of the recorded neurons were covered. In the V1 experiments, the stimulus was $160 \times 160$ pixels (4.5° visual angle) in size for monkey A, and $250 \times 250$ pixels (6.9° visual

angle) for monkey I, also covered all receptive fields but did not overlap with the central fixation point.

The three stimulus categories comprised images of natural scenes, morphed images, and scrambled images. Each category consisted of three image samples. The original stimulus images cannot be presented in the paper due to the journal's license policy. The natural images were corrected to have the same average pixel intensity as the gray background and equal contrast. These images served as basis for the morphed and the scrambled images, respectively. The morphed images were created with the diffeomorphic transformations developed by Stojanoski and Cusack[53]. Using the MATLAB function published by the authors, the following parameters were selected: maxdistortion = 80 and nsteps = 20. The outcome at the end stage of the transformation (corresponding to maximally deformed images) was taken as the morphed stimuli in this experiment. This manipulation destroys higher-order semantic information and preserves basic perceptual Gestalt criteria. Scrambled images were created by adding uniform random noise to the phase spectrum of the original natural images while preserving the amplitude spectrum.

## Data processing

Data acquisition was performed using the TDT system (Tucker-Davis Technologies, Alachua, Florida, USA). Signals were amplified and digitized at about 24.4 kHz (TDT PZ2/PZ5 Preamplifier). In the V4 experiments, the original raw signal was passed through a zero-phase 4th order Butterworth filter with passbands between 500 to 3000 Hz to extract activity in spiking frequency band. From this filtered time series, any event crossing the negative threshold of four times the estimated noise level was identified as multi-unit spiking activity (MUA). The noise level was estimated using the method described in Quiroga et al.[139]. This was done for each channel separately. For the V1 experiments, MUA was detected using the online spike detection algorithm provided by the OpenEx software (Tucker-Davis Technologies, Alachua, Florida, USA). The spike threshold was set at four times the standard deviation of the filtered spiking band activity (400–3000 Hz). Only MUA was analyzed in this study. The trials of all conditions were aligned to stimulus onset (time zero). Data were analyzed within a time window ranging from −500 ms to 1800 ms in the V4 experiments, and from −1000 ms to 2000 ms in the V1 experiments.

To estimate the single trial peak response latency, we used a binning window of 30 ms and moved the window in steps of 1 ms. This window size was selected to slightly "smooth" the rate estimation at the single-trial level. Using smaller step sizes would increase the accuracy of peak time estimation but also increase the scatter. For the decoding analysis based on firing rate (spike count), we used a binning window of 100 ms and a step size of 100 ms. This longer counting window was selected to allow for longer integration time and better classification performance.

All decoding analyses (except rank order decoding, see below) were based on naive Bayes classifiers, provided by the MATLAB Statistics and Machine Learning Toolbox. To test for stimulus-specificity, measurements from each channel were treated as independent variables, i.e., predictors, with repeated measurements across trials. Image identity was used as class label. For response time decoding, the predictors were vectors of multi-channel response onset or peak latencies; for firing rate decoding, the predictors were vectors of multi-channel firing rates, and independent decoders were trained at successive time windows. To examine how fast stimulus-specific information can be read out from the earliest responding units, or how many fastest responding units were needed to decode stimulus identity, we first determined the average response latency for each channel (over trials and stimuli), gradually included more and more channels with shortest latencies as predictors, and used the response latencies of the included channels to decode stimulus identity. This allowed us to evaluate the

changes in decoding accuracy as we increased the number of fastest responding channels. To compare these results based on response latencies to the decoding performance based on firing rates, we changed the input to the classifiers from latency values across channels to firing rates summed in the interval between stimulus onset and the previously determined latencies that were used in the latency decoding analysis. Channels and latency intervals were kept the same. In all cases, a 10-fold cross validation was used, and the variation of classification accuracy over the 10-fold repetitions was used to estimate the confidence interval of the classifier.

For decoding stimulus identity based on response rank order across channels, we adopted a template matching procedure. Specifically, during "training", we obtained a template rank order for each stimulus based on the trial-averaged responses. During "testing", we compared the rank order obtained from single-trial responses with all the templates, and used the stimulus label associated with the most similar template as predicted outcome for each trial. As measurement of similarity, we used Spearman's rank order correlation.

Principal component analysis was applied to visualize the high-dimensional trajectories of the firing rate vectors (window size 30 ms, step size 1 ms). Specifically, calculations were based on the assumption that the native space was spanned by the multi-dimensional firing rate, each channel being treated as an independent dimension. The single-trial firing rate time series across ramp, intensity, and stimulus category conditions were concatenated to form a channel-by-(time × condition) matrix. PCA performed on the covariance matrix of this data matrix yielded the eigenvectors, among which the ones associated with the top three largest eigenvalues were kept. The resulting channel-by-3 matrix was used to project all firing rate data points into the 3-dimensional principal component space. Single-trial projections in the same condition were averaged for visualization of the trajectories. Also shown in the figures were the meshes representing covariance structures. These were produced by transforming a unit sphere by a square-rooted (in matrix sense) covariance matrix scaled by a factor of 0.5, and translated to the designated locations, e.g., baseline or endpoint. All calculations of distances were performed in the native space without PC projection or dimensionality reduction, assuming Euclidian geometry.

## Network simulation

We simulated a recurrent spiking neural network of N = 250 neurons where the ratio between excitatory and inhibitory neurons was 4:1 (Ne = 200, Ni = 50, custom-written simulation in MATLAB). We used the Izhikevich model to simulate single neurons, whose parameters were the same as in ref. 140. The excitatory cells were modeled as regular spiking cells (RS. $(a, b) = (0.02, 0.2)$. $(c, d) = (−65, 8) + (15, −6)*r^2$ where r is random variable of standard uniform distribution) and inhibitory neurons as fast spiking cells (FS. $(a, b) = (0.02, 0.25) + (0.08, −0.05)*r^2$. $(c, d) = (−65, 2))$, each with random heterogeneity. All neurons were randomly connected with a connection probability of 40%. The connection strength was initialized to be uniformly distributed between [0, 1]. To simulate external input current, we used MNIST digit images scaled to the size of 7-by-7, and mapped to a subset of 49 excitatory neurons. In addition, all neurons received random background noise as external input.

During training, the connection strengths of the recurrent connections were modified through a multiplicative spike timing-dependent plasticity (STDP) rule as implemented in refs. 55,56,141. The time constants for potentiation and depression were both set to 20 ms, and the learning rates to 0.02. Connection strengths were allowed to change between (0, 2]. Only excitatory connections were subject to plastic modification. Each simulated training trial consisted of a 100 ms baseline period and 50 ms stimulation period when the stimulus was turned on (without ramp in training). The network weights would stabilize after roughly 200 trials.

**Article** https://doi.org/10.1038/s41467-023-38587-2

During testing, the connection strengths were fixed, and various ramp conditions were simulated. In all conditions, the simulated trials consisted of a 100 ms baseline period, various ramp periods of 0 ms (no ramp), 50 ms (fast ramp) or 100 ms (slow ramp), and a plateau period of 50 ms, all conditions being qualitatively similar to the experimental trial structures. Spikes from excitatory neurons were recorded and later analyzed.

## Reporting summary
Further information on research design is available in the Nature Portfolio Reporting Summary linked to this article.

## Data availability
The data that support the findings of this study are available from the corresponding author upon reasonable request. Source data are provided with this paper.

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

## Acknowledgements

We thank Richard Saunders and Michael Schmid for planning and performing the surgical implants for the V4 experiments. We are grateful to Rasmus Roese and Eleni Psarou for assisting in experiments. We thank Pascal Fries for sharing his laboratory for performing some of the experiments. Martin Vinck and Jennifer Goldman have provided helpful comments on the manuscript. This work was supported by the Ernst Strüngmann Foundation, Max Planck Society, the Human Frontier Science Program (HFSP RGP0044/2018), and the Deutsche Forschungsgemeinschaft (DFG Reinhart Koselleck Project 325248489).

## Author contributions

Experimental design: Y.Y.L., K.S., and A.P. Experimentation: Y.Y.L., K.S., and J.K.L. Data analysis: Y.Y.L. and A.L. Simulation: Y.Y.L. and H.X.H. Writing—original draft: Y.Y.L. and W.S. Writing—review and editing: all authors. Supervision: W.S.

## Funding

## Competing interests

The authors declare no competing interests.
