## [Peer Review File · Nature Communications]

Robust Encoding of Natural Stimuli by Neuronal Response Sequences in Monkey Visual CortexREVIEWER COMMENTS

Reviewer #1 (Remarks to the Author):

In their exciting study, Yiling Yang and colleagues examine population responses in monkey visual cortex. They find that responses can be ordered sequentially and that these sequences across variants of the task in which the latency and absolute intensities are parametrically manipulated. The fact that the dilation factor of the sequences did not change linearly with ramp duration is used to support the model that sequences may be the result of 'priors' or existent cortical wiring. Consistent with the involvement of recurrent cortical networks shaped by experience, the authors also found that normal stimuli resulted in faster and more robust responses than morphed or scrambled images. They also perform a simple 'proof of concept' recurrent network model and find that STDP leads to stimulus-specific response sequences.

Overall, I really liked this paper, and I can only suggest a few minor tweaks that can help to improve upon this work. It is well-written, broadly important, and timely – and I think it should be published in Nature Communications without much further modification.

SUGGESTIONS:

It would be helpful to have more information concerning the data sets used for each figure. How many neurons were analyzed? How many sessions? Where were these neurons recorded? On this last point, I had assumed that Figures 1 & 2 were performed in V1, but the statement on Ln. 312 ('because these results were obtained in a higher visual area') suggests otherwise.

I would appreciate if population data from multiple single trials (maybe 10?) were presented to better appreciate the stability of the rank order of neurons across trials. I was envisioning something like an experimental version of Figure 4B.

The result about the persistence of the responses seems only peripherally related to the rest of the narrative. The authors should either try to integrate this section into the abstract and the discussion or drop it altogether. More broadly, it seems fitting that the story ends with the model (as is the case in the abstract).

The affiliation numbers for the authors do not appear to be correct.

A few relevant papers should be cited here: Rajan et al., Neuron, 2016 (a highly influential review on recurrent models and sequences); Kim et al., PNAS, 2016 (a demonstration that sequences can be experimentally established with experience); Elmaleh et al., Neuron, 2021 (a dissection of local and long-range influences on sequence generation in the songbird).

Ln. 184 – is Figure 1f the correct one to reference here?

Figure 1 – the gray tones for low and high intensity are very difficult to distinguish.

Ln. 234 – should be 'led' not 'lead'

Ln. 304 – why is the decoding accuracy higher for the fast ramp than the slow ramp condition?

Ln. 310 – authors should measure (if possible) whether the unnatural stimuli result in a less precisely timed sequence

The PCA plots in Figure 6A are difficult to parse. Can the three conditions be plotted separately?

Reviewer #2 (Remarks to the Author):

The basic ideas behind this paper are important and timely. They are right to point out that the vast majority of work in artificial neural networks uses models that have no interesting dynamics – processing is typically limited to feedforward propagation of floating-point numbers that are supposed to correspond to the firing rates of neurons. A seven-layer network such as AlexNet will process an image in exactly 7 (massively parallel) steps. And there is nothing of interest to learn from the dynamics.

Some labs, including DiCarlo's lab at MIT have reported that, unlike artificial networks, real networks show interesting dynamics that would not be explained by purely feedforward mechanisms. And they have argued that this is because of recurrent connectivity.

In the current paper, the authors make a similar claim. They describe interesting and reliable sequences of activation to both flashed images and images that are progressively ramped on either relatively quickly (over 500 ms) or more slowly (over 1 second). The data is certainly very interesting, and it is true that very few researchers have bothered to look for such phenomena.

My main problem with the paper as it stands is that they use these reliable sequences to argue that this can only be explained by recurrent connectivity within the cortex – much as DiCarlo has done.

But the problem is that even a simple feedforward spiking circuit will show reliable latency differences depending on physical characteristics of the stimulus. Remarkably, such latency variations were visible in the very first recordings of spiking activity from the optic nerve by Lord Adrian in 1927, but ignored by the vast majority of neurophysiologists for decades. However, a paper by Gollisch and Meister (Science, 2008) showed that the relative timing of spikes in retinal ganglion cells can be reliably used to transmit information. Contrast sensitive variations in latency are accepted as the explanation of the well-known Pulfrich Pendulum phenomenon – lowering contrast increases latency. Unless the authors think that this requires recurrent connections in the retina, I presume that they would have to agree that their claim in the discussion that there is “no experimental evidence for feedforward mechanisms capable of generating reliable sequence-based information” is simply not true.

For me, it seems clear that, even without any learning, and without any recurrent connections, you would expect that there would be reliable information in the relative order of firing across neurons. The authors admit that in the auditory system, temporal information is used (for sound localization), so it would be surprising if such phenomena were not also true for vision. And they cite a large number of studies in several sensory modalities that demonstrate temporal information.

The authors' belief that temporal sequences imply recurrent processing is made clear in the final conclusion where they argue that "the cerebral cortex complements the feedforward processing of rate-coded information by exploiting the dynamics of recurrent networks to generate a temporal code...". Do they really believe that a feedforward spiking network has no dynamics?

So, while the authors have done a great service to the community by looking in detail at the information that is contained in temporal dynamics, they overstep the mark by arguing that their findings can only be explained by recurrent processing.

To be clear, recurrent connections may indeed be very important and interesting to study, and those connections could indeed reflect learned information about stimuli. But even feedforward networks can learn. For example, Masquelier and Thorpe (PLoS 2007) showed that neurons equipped with STDP can become selective to faces and motorcycles even in purely feedforward networks. Furthermore, STDP leads to neurons responding faster and more reliably with repeated stimulation – exactly as they present authors note. But, again, no need for recurrent architectures.

The analysis based on the differences in the precise sequence with the three different types of stimulus was weak. Yes, they can rule out a simple model where a feedforward network of neurons with thresholds for firing that are fixed at a particular value of physical contrast. That is clearly not a good model. But the time to reach threshold for a leaky integrate and fire neuron depends on many factors. Changing the leakage will increase the time taken to reach threshold and produce all sorts of non-linearities that could also be consistent with those changes.

Overall, I found the conclusion that the "temporal sequences observed in the experiments resulted from interactions in a recurrent network in which the strengths of the coupling connections had been shaped by experience" too strong. The fact is that reliable temporal sequences to flashed or ramped stimuli would be obtained even in a feedforward neural network model with no recurrent connections. The main reason that such effects are not observed in the vast majority of current neural network models is that those models typically do not have spikes – they send floating point numbers that supposedly represent the neuron's firing rate instantaneously, totally removing the dynamics that immediately become interesting as soon as you start to take spiking seriously. Interesting dynamics, like the ones shown here, are not only limited to recurrent networks.

All this is rather regrettable, because as I stated, I think that this is a really interesting and original study. If the authors were to accept that feedforward networks can also have interesting dynamics, then I would be much happier to recommend publication.

I have a number of other comments.

Firstly, it was very unclear whether any of the channels corresponded to single neurons, or whether all the recordings were multiunit. If they can distinguish between these options, it would be interesting to see whether a small number of single units (rather than multiunit channels) could allow even more precise decoding. Indeed, to make the point, if you were to record from just 4 individual cells and looked at the ordering of firing, you would have 4! (24) different orders. In principle, those four cells could classify up to 24 different images.

Given the power of rank order coding, it is really unfortunate that the data set was collected with just 3 images – well below the number that could be decoded with just four neurons.

Using a small number of repeated images also makes it highly likely that the visual system could have learned about the statistics of those particular images, and this makes it harder to make conclusions about how the system would respond to unpredictable images. It would have been useful to have a condition where random images (selected from the ImageNet database for example) were shown. Would such images produce more (or perhaps less) neural activity than the highly predictable images used in the study? Would the authors predict that only the familiar stimuli would show interesting sequences? If they authors have any such data, it would greatly enhance the value of the data. My bet is that even a totally novel natural image would show clear and reliable sequences of activation. It might even show better reliability than scrambled images (as demonstrated here) but with no need for learning at all.

The use of just 3 stimuli also markedly reduces the power of the analysis. The decoder never has to choose between more than 3 stimuli and makes judgment in favor of one of the stimuli even when nothing is present. I would have preferred an analysis where the system had to initiate a three-choice response when there was enough information to conclude that a stimulus had been presented, and not simply choose which of the three stimuli is the most likely at every instant.

In the modelling study, I was puzzled by the statement that “Already at the beginning of training the networks produced sequences but that they were highly variable”. What would cause that variability? Was noise added to the activity? I would have thought that even the simplest neural model of center-

surround receptive fields in the retina would have latencies that would vary depending on the stimulus because higher local contrast lead to shorter latencies.

I was also surprised that the modelling was done with a relatively boring image of a digit. Why not use the natural images used for the study? And was the modelling done with a spiking neural network simulator? Or a conventional network that represents firing rates with floating point numbers? These are important issues that are currently unclear from the presentation.

The paper refers to values for the total spikes per second that are several thousands. This is a strange way to talk about the data, and is almost meaningless, because it is presumably calculated for an undefined and large number of channels and where is it unclear whether the channels refer to individual neurons or multiunit activity.

The authors also spend a lot of time discussing the information about peak latency for different channels. Indeed, much of the data in support of reliable sequences is built on the assumption that information about the peak latency in different channels could be used in the brain. It may be trivial for the experimenter to measure the latency of the peak. But firstly, this makes the basic assumption that rates are critical, whereas the whole point of rank order coding is to get away from rates. Secondly, I know of no plausible neurophysiological mechanism that could respond selectively to a peak at a particular latency. In my opinion, all that section could be dropped with no real loss. Onset latency variations across populations of neurons are by contrast intrinsically much more interesting, as well as being much faster to compute – as the authors point out.

So, in conclusion, I must admit that while interesting, the paper reaches conclusions that are not supported by the data.

**AUTHOR REMARK**

We wish to thank our reviewers for their sacrifice of time, their scrutiny and the many constructive
suggestions that we now tried to incorporate into the revised version of our manuscript. Below we
address the reviewers' comments (in black) point by point, mark our responses in green, and
copy/paste here the corresponding revisions from the updated manuscript in blue. The revisions are
also highlighted in the updated manuscript.

**REVIEWER COMMENTS**

Reviewer #1 (Remarks to the Author):

In their exciting study, Yiling Yang and colleagues examine population responses in monkey visual
cortex. They find that responses can be ordered sequentially and that these sequences across
variants of the task in which the latency and absolute intensities are parametrically manipulated.
The fact that the dilation factor of the sequences did not change linearly with ramp duration is used
to support the model that sequences may be the result of 'priors' or existent cortical wiring.
Consistent with the involvement of recurrent cortical networks shaped by experience, the authors
also found that normal stimuli resulted in faster and more robust responses than morphed or
scrambled images. They also perform a simple 'proof of concept' recurrent network model and find
that STDP leads to stimulus-specific response sequences.

Overall, I really liked this paper, and I can only suggest a few minor tweaks that can help to improve
upon this work. It is well-written, broadly important, and timely – and I think it should be published
in Nature Communications without much further modification.

We thank the reviewer for examining our manuscript and for his/her encouraging comments.

SUGGESTIONS:

It would be helpful to have more information concerning the data sets used for each figure. How
many neurons were analyzed? How many sessions? Where were these neurons recorded? On this
last point, I had assumed that Figures 1 & 2 were performed in V1, but the statement on Ln. 312
('because these results were obtained in a higher visual area') suggests otherwise.

We thank the reviewer for drawing our attention to this omission of important information. Indeed,
we have forgotten to mention that all the results presented in the main text were obtained from
area V4, and that all V1 results, which were very similar, are shown in the supplementary materials.
Initially, the experiment was planned for area V4 but later we had the opportunity to extend and
confirm the findings with experiments in V1 of two additional monkeys. V4 and V1 data were
collected with chronically implanted 64- and 32-channel devices, respectively. We analysed all the
channels (multi-unit activity) without sorting single neurons due to signal-to-noise constraints. V4
experiments comprised in total 12 sessions (6 sessions per animal), and V1 experiments comprised 6
sessions (3 per animal). This information and more essential details of the dataset have now been
added in the appropriate paragraphs of the Results and Methods sections, and are also pasted
below.

L128 (line number 128 in the revised manuscript, same convention below): Four awake macaque
monkeys were presented with these stimuli in a passive viewing task and multi-unit activity (MUA)
was recorded from visual area V4 in two monkeys with a 64-channel Utah array (Blackrock
Microsystem, Salt Lake City, Utah, USA. Supplementary Figure 1) and from area V1 in another two

monkeys with a 32-channel Microdrive (Gray Matter Research, Bozeman, Montana, USA). We did
not sort for single units and analyzed only MUA in this study. We present the findings obtained from
area V4 in the main text and refer readers to the Supplementary Information for the results from
area V1.

L766: In total, monkey H performed 2955 trials in 6 sessions (492.5 ± 113.3 (s.d.) trials per session, or
159.8 ± 25.1 trials per condition); monkey K performed 4679 trials in 6 sessions (779.8 ± 209.9 trials
51 per session, or 253.3 ± 48.4 trials per condition).

L778: In total, monkey A performed 3133 trials in 3 sessions (1044.3 ± 52.7 trials per session, or
1044.3 ± 1.1 trials per condition); monkey I performed 2418 trials in 3 sessions (806.0 ± 391.6 trials
54 per session, or 806.0 ± 2.0 trials per condition).

L810: Only MUA was analysed in this study.

I would appreciate if population data from multiple single trials (maybe 10?) were presented to
better appreciate the stability of the rank order of neurons across trials. I was envisioning something
like an experimental version of Figure 4B.

We agree with the reviewer and are generally in favour of presenting raw single-trial data. We had
inspected our raster plots and single-trial histograms prior to averaging but realized that the single-
trial spike data did not appear visually as striking and intuitive as the averaged data (Figure 1c&d) or
simulated data (Figure 4b) because of the inherent variability of responses in awake animals
(fluctuations of baseline activity, different response profiles and signal-to-noise levels across
channels). We therefore decided to show averaged firing rate heat maps, as is commonly done in
the literature to visualize sequences, and to complement these qualitative representations with
extensive statistical quantification of the characteristic features of sequences.

The result about the persistence of the responses seems only peripherally related to the rest of the
narrative. The authors should either try to integrate this section into the abstract and the discussion
or drop it altogether. More broadly, it seems fitting that the story ends with the model (as is the case
in the abstract).

We agree and could consider separating the results on persistence (V1 and V4) into a shorter paper
or moving them into the supplement if the reviewers and the editor suggest doing so. However,
these results provide additional support for the complementarity of rate and temporal codes, for the
Bayesian matching operations and the dynamics of intracortical processing. Moreover, we needed to
analyse rate codes during the initial transient for comparison with the rank code and would have to
find a good reason why we limited these analyses to the phasic response components and did not
include the sustained phase as well. We added sentences to the abstract and to the results to
strengthen the conceptual link between the two sets of results.

L33: Support for such a matching process comes from the additional finding that stimulus-specific
information persists longer in responses to natural than manipulated stimuli.

L285: Therefore, we expected that both the sequences and the rate vectors (see sections below)
might reflect not only the structure of the stimuli but also the extent to which sensory evidence
matched the priors stored in the architecture of the cortical networks.

The affiliation numbers for the authors do not appear to be correct.

Corrected.

Yang Yiling^{1,2,6}, Katharine Shapcott^{1,3}, Alina Peter^{1,2,6}, Huang Xuhui⁴, Andreea Lazar¹, Wolf Singer^{1,3,5*}

¹Ernst Strüngmann Institute (ESI) for Neuroscience in Cooperation with Max Planck Society,
Deutschordenstraße 46, 60528 Frankfurt am Main, Germany

²Faculty of Biological Sciences, Goethe-University Frankfurt am Main, Max-von-Laue-Str. 9, 60438
Frankfurt am Main, Germany

³Frankfurt Institute for Advanced Studies, Ruth-Moufang-Str. 1, 60438 Frankfurt am Main, Germany

⁴Institute of Automation, Chinese Academy of Sciences, Zhongguancun East Road 95, 100190 Beijing,
China

⁵Max Planck Institute for Brain Research, and ⁶International Max Planck Research School (IMPRS) for
Neural Circuits, Max-von-Laue-Str. 4, 60438 Frankfurt am Main, Germany

A few relevant papers should be cited here: Rajan et al., Neuron, 2016 (a highly influential review on
recurrent models and sequences); Kim et al., PNAS, 2016 (a demonstration that sequences can be
experimentally established with experience); Elmaleh et al., Neuron, 2021 (a dissection of local and
long-range influences on sequence generation in the songbird).

The suggested citations have now been inserted in appropriate places.

L66: The dynamics of recurrent neural networks (RNNs) are exploited for computations in artificial
RNNs (Buonomano and Maass, 2009; Jaeger and Haas, 2004; Lazar et al., 2021; Lazar et al., 2009;
Maass et al., 2002; Rajan et al., 2016; Romera et al., 2018) but it is less clear to which extent
biological RNNs capitalize on their dynamics to achieve specific functions (for review see Muller et al.
(2018); Singer (2021)).

L646: As detailed above, the reciprocal connections between columns responding to features which
have a high probability of being correlated in natural environments get strengthened through a
Hebbian mechanism (Bosking et al., 1997; Galuske et al., 2019; Gilbert and Wiesel, 1989; Iacaruso et
al., 2017; Kim et al., 2016; Löwel and Singer, 1992).

L564: Evidence that the timing of discharges matters in neuronal processing is available from studies
in various systems: vision (Bruno and Sakmann, 2006; Burr and Ross, 1979; Delorme, 2003; Gawne et
al., 1996; Gollisch and Meister, 2008; Gray et al., 1989; Havenith et al., 2011; Warzecha and
Egelhaaf, 2000), audition (Carr and Konishi, 1990; Overholt et al., 1992), olfaction (Chong et al.,
2020; Haddad et al., 2013; Junek et al., 2010; Smear et al., 2011; Spors and Grinvald, 2002; Spors et
al., 2006; Wehr and Laurent, 1996; Wilson et al., 2017), somatosensation (Johansson and Birznieks,
2004; Montemurro et al., 2007; Panzeri et al., 2001; Petersen et al., 2002), spatial navigation
(O'Keefe and Recce, 1993; Pastalkova et al., 2008; Skaggs et al., 1996; Wilson and McNaughton,
1994), and motor control (Daou and Margoliash, 2020; Egger et al., 2020; Elmaleh et al., 2021;
Hahnloser et al., 2002; Yu and Margoliash, 1996).

Ln. 184 – is Figure 1f the correct one to reference here?

It was indeed not correct and is now corrected to Figure 1g.

Figure 1 – the gray tones for low and high intensity are very difficult to distinguish.

Indeed. We have now highlighted the contrast between the two intensity conditions in Figure 1e:

And Supplementary Figure 5:

Ln. 234 – should be ‘led’ not ‘lead’

Corrected.

L242: It should be emphasized that even the no ramp condition led to decodable response sequences.

Ln. 304 – why is the decoding accuracy higher for the fast ramp than the slow ramp condition?

We had also expected that the decoding accuracy should improve with ramp duration. However,
 although the ramping diluted the tightly packed latencies of responses to sudden stimulus onset,
 there is apparently a limit to the benefits of ramping. This can have two reasons. First, the very
 gradual increase of stimulus intensity in the slow ramp condition may have limited the number of
 active neurons able to contribute to decoding. Despite increasing the ramp duration, most neurons
 still responded very early (c.f. Figure 1f&g, which underlines the nonlinear relationship between
 ramp duration and the temporal span of sequences). By the time the sequences were complete and
 recognized by the decoder, stimulus intensity was still low and this likely enhanced the variability
 and latency scatter of responses and may even have reduced the number of effectively contributing
 neurons. Second, in the slow ramp condition, the slowly rising firing rates are bound to impede the

ability of our algorithm to precisely determine the timing of responses. This increased variability
likely also accounts for reduced decodability of slow ramp responses. Thus, there seems to be a
trade-off between speed and accuracy. Had time and animal resources permitted, we would have
liked to vary the ramp duration more systematically, in order to get a better estimation of the
nonlinearity between ramp duration and neuronal responses.

We think this important question highlights the nonlinearity of neuronal responses and may also be
of interest to other reader. Therefore we inserted our explanation into the Discussion section (L530).

Ln. 310 – authors should measure (if possible) whether the unnatural stimuli result in a less precisely
timed sequence

We agree. We have tried to measure the variance of response timing of *individual channels/neurons*,
pooled the variance across channels and compared the pooled variances of responses to the
different stimulus categories. However, this measure, which is a rather insensitive metric as the
reviewer has probably suspected, did not yield statistically significant differences. One probably has
to develop more dedicated and elaborate measurements to evaluate the precision of response
*sequences*, rather than the precision of single neuron response latencies.

The PCA plots in Figure 6A are difficult to parse. Can the three conditions be plotted separately?

We have added vertical offsets between trajectories of the three conditions.

Figure 6a:

L465 (caption of Figure 6): Within each panel, trajectories of the three stimulus categories have been
 displaced vertically to reduce visual cluttering; otherwise the three baseline covariance meshes
 (black) should overlap.

Supplementary Figure 18a:

Reviewer #2 (Remarks to the Author):

The basic ideas behind this paper are important and timely. They are right to point out that the vast
 majority of work in artificial neural networks uses models that have no interesting dynamics –
 processing is typically limited to feedforward propagation of floating-point numbers that are
 supposed to correspond to the firing rates of neurons. A seven-layer network such as AlexNet will
 process an image in exactly 7 (massively parallel) steps. And there is nothing of interest to learn from
 the dynamics.

Some labs, including DiCarlo's lab at MIT have reported that, unlike artificial networks, real networks
show interesting dynamics that would not be explained by purely feedforward mechanisms. And
they have argued that this is because of recurrent connectivity.

In the current paper, the authors make a similar claim. They describe interesting and reliable
sequences of activation to both flashed images and images that are progressively ramped on either
relatively quickly (over 500 ms) or more slowly (over 1 second). The data is certainly very interesting,
and it is true that very few researchers have bothered to look for such phenomena.

My main problem with the paper as it stands is that they use these reliable sequences to argue that
this can only be explained by recurrent connectivity within the cortex – much as DiCarlo has done.

We thank the reviewer for examining our manuscript and for sharing his/her comments on and
interpretations of our findings. We would like to make a few general remarks before responding to
the reviewer's comments point by point.

After we had carefully studied the reviewer's comments, we feel that there is more agreement than
dissent between the reviewer's and our interpretations with respect to the significance of our main
findings: the stimulus specificity of response sequences, the option for fast read out of these
sequences and their dependence on natural image statistics. However, we have certainly failed to
consider in sufficient depth other mechanisms than recurrent processing for the generation of
response sequences. In revising the manuscript we attempted to give more space to alternative
hypotheses. Therefore, our responses to the reviewer's comments should not be seen as a rebuttal
but rather as an attempt to discuss the pro and cons of different interpretations.

One of the reviewer's major concerns is that feedforward rather than recurrent networks could
account for our findings. We have considered the possibility that the sequences were actually the
consequence of feedforward mechanisms and have designed a number of control experiments to
test this hypothesis but the result was, that simple versions of feedforward mechanisms fell short of
explaining some features of the observed sequences. These features were the following. First, the
preservation of sequence order despite changes in absolute latencies and despite changes in relative
latencies that occurred with the compression and dilation of sequences. Second, the stability of
sequences despite unavoidable changes in afferent drive caused by microsaccades. The fixation
window allowed the monkey to make small eye movements and this likely had the effect that the
correspondence between the contours of the stimuli and the neurons' receptive fields changed
within a trial and between trials, likely causing variations of response latencies all along the
feedforward transmission chain from the retina to V4. Third, the dependence of the persistence of
the decodable rate responses on natural image statistics. We felt that these observations were
better explained by reverberation in a recurrent network than by serial forward propagation of
activity. Finally, a very simple and generic recurrent network model endowed with Hebbian synapses
reproduced at least qualitatively most of our experimental findings without much parameter
tweaking. Our control experiments do of course not exclude that a feedforward network endowed
with a combination of complex nonlinear and hitherto unidentified transfer functions would be able
to reproduce the highly adaptive dynamics revealed by our results. However, applying Occam's razor
we opted for the most straight forward interpretation. We did not intend to rule out that specifically
crafted feedforward models could reproduce some of our findings. This possibility is definitely worth
further systematic investigation and confirmation.

We are well aware of the fact that concepts on visual processing emphasize serial feedforward
processing in hierarchical architectures and that this notion is further supported by the success of
deep neural networks that share this strategy. However, anatomical evidence indicates that in
biological neural systems recurrent connections within and between the different areas of the
processing hierarchy outnumber by far the feedforward connections. Furthermore, biological
systems exhibit exceedingly complex dynamics that are absent in feedforward networks and can
only be accounted for by recurrent processing. Accordingly, concepts emphasizing the importance of
recurrence and dynamics are readily accepted in other fields of neuroscience. They serve as
framework for studies on other systems of the vertebrate brain (e.g., other sensory systems,
hippocampus and motor cortex) and for investigations of nervous systems of other species (e.g.,
insects, worms and crustaceans). Because of the striking similarities between cortical areas, it seems
plausible to assume that the visual system also relies on a combination of feedforward and recurrent
processing.

But the problem is that even a simple feedforward spiking circuit will show reliable latency
differences depending on physical characteristics of the stimulus. Remarkably, such latency
variations were visible in the very first recordings of spiking activity from the optic nerve by Lord
Adrian in 1927, but ignored by the vast majority of neurophysiologists for decades. However, a
paper by Gollisch and Meister (Science, 2008) showed that the relative timing of spikes in retinal
ganglion cells can be reliably used to transmit information. Contrast sensitive variations in latency
are accepted as the explanation of the well-known Pulfrich Pendulum phenomenon – lowering
contrast increases latency. Unless the authors think that this requires recurrent connections in the
retina, I presume that they would have to agree that their claim in the discussion that there is “no
experimental evidence for feedforward mechanisms capable of generating reliable sequence-based
information” is simply not true.

We are very much in agreement with the reviewer that spike latencies can transmit information, and
have indeed cited the work of Gollisch and Meister (Science, 2008) to support this notion. We also
agree with the reviewer that physical characteristics of a stimulus like contrast can change spike
latencies. Besides the original work by Lord Edgar Adrian, we would like to mention that a few years
later Haldan Hartline also reported variations in optic nerve discharge latency in response to varying
stimulus intensities (e.g., Hartline & Graham, 1932; Hartline 1938). In later related studies in the
Limulus eye Hartline and Ratliff actually stated that an image stimulus could be reconstructed
perfectly well from transforming absolute response latencies into luminance values. The authors
mentioned that there were later response components (after-discharges) whose amplitude also
signalled stimulus intensity and allowed image reconstruction. Interestingly, since then, the field has
concentrated mainly on these "after-discharges". The stability of the retinal sequences, that are
defined by absolute response latencies, does of course require that the contours of the stimulus
always fall on exactly the same retinal loci - which was likely not the case in our experiments.

In our experiments, the texture of the stimuli contains high spatial frequencies and hence the
luminance of the stimuli changes substantially over short distances. Because the animals perform
microsaccades while they fixate, the luminance of the contours in the receptive fields is constantly
changing. Therefore it is surprising that the sequences maintained the same rank order.
Furthermore, the contrast differences between the *same* stimuli presented with different ramp
conditions were much larger than the contrast differences between the *different* stimuli due to the
low-contrast regime of our paradigm. Thus, if contrast played a major role, the correlation between
onset latencies would be large between ramp conditions and small between stimuli, which is not the

case (c.f. Figure 1, Supplementary Figure 2 and Supplementary Figure 3). Variations in stimulus
contrast also cannot explain the differences in stimulus specificity of response sequences between
different types of stimuli (natural, morphed and scrambled), because stimulus onset kinetics were
identical for all stimulus types. We have obviously failed to make clear that contrast dependent
variations in latencies can *alone* not account for *all* the features of our response sequences
(invariance to temporal compression and dilation, stability despite changing retinal correspondence,
dependence on priors about natural image statistics).

We must have been imprecise in our writing because the reviewer assumes that we said "there is no
experimental evidence for feedforward mechanisms capable of generating reliable sequence-based
information" (last sentence of the reviewer's comments above). However, we stated in our
discussion that "there are yet no experimental data in support of feedforward mechanism capable of
generating response sequences, whose sequence order is invariant to temporal compression and
dilation". We apologize for not having conveyed our position with the required clarity.

For me, it seems clear that, even without any learning, and without any recurrent connections, you
would expect that there would be reliable information in the relative order of firing across neurons.
The authors admit that in the auditory system, temporal information is used (for sound localization),
so it would be surprising if such phenomena were not also true for vision. And they cite a large
number of studies in several sensory modalities that demonstrate temporal information.

We agree with the reviewer that temporal information is used in many neural systems, as we have
cited extensively. To the best of our knowledge, in other systems such as hippocampus, motor
cortex, and auditory systems, it is usually assumed and even accepted that such temporal dynamics
in neuronal responses emerged from recurrent networks. We cannot think of a good argument why
such mechanisms should not hold in visual systems.

The authors believe that temporal sequences imply recurrent processing is made clear in the final
conclusion where they argue that "the cerebral cortex complements the feedforward processing of
rate-coded information by exploiting the dynamics of recurrent networks to generate a temporal
code...". Do they really believe that a feedforward spiking network has no dynamics?

Again, we apologize for having been unclear in our writing. We do of course not deny that
feedforward networks can produce sequences. As demonstrated very early on by Abeles in his work
on synfire chains, strictly feedforward networks do generate informative sequences but to the best
of our knowledge these sequences are not compressible and also lack the rich self-organizing
dynamics observed in natural neuronal networks that are required to generate temporal codes.

So, while the authors have done a great service to the community by looking in detail at the
information that is contained in temporal dynamics, they overstep the mark by arguing that their
findings can only be explained by recurrent processing.

We have revised the manuscript to leave more room for alternative interpretations and emphasize
the feedforward option at the very end of our concluding remarks.

L718: We have proposed recurrent processing as the most parsimonious explanation for the
generation of response sequences and rank order codes. However, this does not exclude that

feedforward processing could have produced the same result. Any recurrent network can be
unrolled in a multilayer feedforward network and with appropriate implementation of
nonlinearities, such feed forward networks can in principle simulate recurrent processes (Kubilius et
al., 2018; Liao and Poggio, 2016; Nayebi et al., 2018; Zamir et al., 2017). Our data do not allow us to
distinguish between these possibilities but emphasize the putative importance of coding strategies
exploiting temporal relations among neuronal responses.

L503: It is of course conceivable that feedforward networks can also reproduce our findings if
endowed with the required mix of nonlinearities. However, to the best of our knowledge, there are
yet no experimental data in support of feedforward mechanisms capable of generating response
sequences, whose sequence order is invariant to temporal compression and dilation. Sequence
generation is an intrinsic property of recurrent networks, and a very simple recurrent spiking
network model could already reproduce many of our findings without extensive parameter tuning or
handcrafting of nonlinearities. Therefore, we consider recurrent processing as the most
parsimonious explanation of our results.

L148: This revealed that the stimulus-specific rank orders of response latencies were by and large
preserved for the fast ramp condition although the absolute latencies had decreased, leading to a
temporal compression of the sequences (diagonal panels in Figure 1d and Supplementary Figure 2b).

L197: This disproportional scaling of sequence span and of the latency distributions of the nodes'
responses suggests that the two variables are not solely determined by stimulus parameters but
depend also on network interactions. Further control analyses (Supplementary Information,
Supplementary Figure 7 and Supplementary Figure 8) and simulation experiments (see below)
support the notion that the sequences do not simply reflect the temporal structure of afferent
signals nor their energy nor different sensitivities of the nodes to stimulus energy. Rather, the
sequences seem to be shaped by or emerge from complex interactions within cortical networks.

To be clear, recurrent connections may indeed be very important and interesting to study, and those
connections could indeed reflect learned information about stimuli. But even feedforward networks
can learn. For example, Masquelier and Thorpe (PLoS 2007) showed that neurons equipped with
STDP can become selective to faces and motorcycles even in purely feedforward networks.
Furthermore, STDP leads to neurons responding faster and more reliably with repeated stimulation
– exactly as they present authors note. But, again, no need for recurrent architectures.

We agree with the reviewer that feedforward networks can learn. Our experiments were also very
much inspired by Simon Thorpe's work, including the paper the reviewer referred to. As we read it,
Masquelier and Thorpe (PLoS, 2007) demonstrated that spike latency can be used to represent visual
features and to perform unsupervised learning via STDP. To train this small proof-of-concept
network, the authors had to implement a winner-take-all strategy in certain layers (specifically,
layers S1 and C1) and k-winner-take-all strategy in another layer (i.e., layer S2), by introducing local
lateral inhibition. Although only required in the learning phase and thus not explicitly illustrated in
the model structure, such lateral inhibition mimics already some recurrent interactions. Moreover,
even this small "feedforward" model required dedicated handcrafting of structures and parameters,
i.e., when to turn on lateral inhibition, which connections should undergo STDP (i.e., layers C1 to S2),
whether to drop the leakage term, where to apply 1-winner-take-all and where to apply a k-winner-
take-all mechanism. The resulting network was highly specialized and dedicated to a specific task. By
contrast, recurrent networks can reproduce similar functions without much parameter tuning, and

remain adaptable for other tasks. Model simplicity and flexibility “biased” us to favour recurrent
mechanisms. We agree that we should have acknowledged that feedforward network structures
could also account for our results if they exploited the universe of nonlinearities and the virtually
inexhaustible combinatorics of diverging and converging connections. Theoretically, any recurrent
network can be unrolled into a feedforward network in which each time step of the reciprocal
interactions is represented by a layer. We mention this now at the end of our discussion. However, if
the recurrent network of a cortical area would have to be unrolled to capture the numerous virtually
simultaneous interactions among the nodes this would require extremely deep networks. Given the
time constants of neuronal processes, the conversion of the parallel computations of recurrent
networks into serial computations in feedforward networks would be hard to reconcile with
processing speed and hardware constraints. Even if we assume that the brain operates digitally at
discrete time steps, it is not trivial to determine how many steps back in time a recurrent network
should be unrolled. Biological neural networks need to perform computations flexibly across a wide
range of timescales, from milliseconds to hours and beyond. One would have to convert a recurrent
network into many, if not infinite, feedforward networks of different depths in order to cope with
tasks of different time spans, significantly sacrificing flexibility.

The analysis based on the differences in the precise sequence with the three different types of
stimulus was weak. Yes, they can rule out a simple model where a feedforward network of neurons
with thresholds for firing that are fixed at a particular value of physical contrast. That is clearly not a
good model. But the time to reach threshold for a leaky integrate and fire neuron depends on many
factors. Changing the leakage will increase the time taken to reach threshold and produce all sorts of
non-linearities that could also be consistent with those changes.

As we have acknowledged, we agree with the reviewer that feedforward networks with carefully
designed nonlinearities could explain our data. More systematic investigations focused on this issue
would be required but our methods do not allow us to perform such analyses. We could only show
that it is not trivial to produce scalable sequences. We agree with the reviewer that leakage
introduces time as coding space. However, in this case the predictions derived from comparisons of
threshold passing with slow vs. fast ramps and low vs. high contrast stimuli would deviate even
further from our measurements.

Overall, I found the conclusion that the “temporal sequences observed in the experiments resulted
from interactions in a recurrent network in which the strengths of the coupling connections had
been shaped by experience” too strong. The fact is that reliable temporal sequences to flashed or
ramped stimuli would be obtained even in a feedforward neural network model with no recurrent
connections. The main reason that such effects are not observed in the vast majority of current
neural network models is that those models typically do not have spikes – they send floating point
numbers that supposedly represent the neuron’s firing rate instantaneously, totally removing the
dynamics that immediately become interesting as soon as you start to take spiking seriously.
Interesting dynamics, like the ones shown here, are not only limited to recurrent networks.
All this is rather regrettable, because as I stated, I think that this is a really interesting and original
study. If the authors were to accept that feedforward networks can also have interesting dynamics,
then I would be much happier to recommend publication.

In the original manuscript, we have acknowledged the possibility of feedforward mechanisms. As the
reviewer suggested, we have now modified the manuscript to further strengthen this alternative
explanation.

L718: We have proposed recurrent processing as the most parsimonious explanation for the
generation of response sequences and rank order codes. However, this does not exclude that
feedforward processing could have produced the same result. Any recurrent network can be
unrolled in a multilayer feedforward network and with appropriate implementation of
nonlinearities, such feed forward networks can in principle simulate recurrent processes (Kubilius et
al., 2018; Liao and Poggio, 2016; Nayebi et al., 2018; Zamir et al., 2017). Our data do not allow us to
distinguish between these possibilities but emphasize the putative importance of coding strategies
exploiting temporal relations among neuronal responses.

L503: It is of course conceivable that feedforward networks can also reproduce our findings if
endowed with the required mix of nonlinearities. However, to the best of our knowledge, there are
yet no experimental data in support of feedforward mechanisms capable of generating response
sequences, whose sequence order is invariant to temporal compression and dilation. Sequence
generation is an intrinsic property of recurrent networks, and a very simple recurrent spiking
network model could already reproduce many of our findings without extensive parameter tuning or
handcrafting of nonlinearities. Therefore, we consider recurrent processing as the most
parsimonious explanation of our results.

I have a number of other comments.

Firstly, it was very unclear whether any of the channels corresponded to single neurons, or whether
all the recordings were multiunit. If they can distinguish between these options, it would be
interesting to see whether a small number of single units (rather than multiunit channels) could
allow even more precise decoding. Indeed, to make the point, if you were to record from just 4
individual cells and looked at the ordering of firing, you would have 4! (24) different orders. In
principle, those four cells could classify up to 24 different images.

We had mentioned that we recorded multi-unit activity (L129 and L806), and now also added
explicitly that we did not sort for single units (L128, L810), mainly because of signal-to-noise
considerations. As the reviewer pointed out, and we fully agree, one of the advantages of rank order
coding is flexible permutation of network nodes, thus providing a virtually infinite coding space.
Although we did not sort single units, we performed an extrapolation by examining to which extent
the subset of earliest responding channels conveyed sufficient information for decoding stimulus
identity (c.f. Figure 2). As predicted, only a handful of channels were needed to decode stimuli, and
the decoding performance increased with more channels. We would predict in agreement with the
reviewer that sorting single units would allow for more precise decoding.

L128: Four awake macaque monkeys were presented with these stimuli in a passive viewing task and
multi-unit activity (MUA) was recorded from visual area V4 in two monkeys with a 64-channel Utah
array (Blackrock Microsystem, Salt Lake City, Utah, USA. Supplementary Figure 1) and from area V1
in another two monkeys with a 32-channel Microdrive (Gray Matter Research, Bozeman, Montana,
USA). We did not sort for single units and analyzed only MUA in this study. We present the findings
obtained from area V4 in the main text and refer readers to the Supplementary Information for the
results from area V1.

L810: Only MUA was analysed in this study.

L245: We then determined how early, on average, each channel (node) started to respond, and
systematically included more and more of the fastest responding channels (Methods). The decoding
accuracy exceeded chance level as soon as more than 4 to 5 channels were included.

Given the power of rank order coding, it is really unfortunate that the data set was collected with
just 3 images – well below the number that could be decoded with just four neurons.

We fully agree and would have loved to collect more data with more stimulus images. For this study,
we required a large number of trials. First, we needed to test numerous conditions (no. images x
intensity levels x ramp durations x stimulus categories). Second, we needed sufficient trials for the
same condition to have enough training and independent test trials for the classifiers. Given the
limited number of trials the monkeys can perform in a session and the constraint to maintain stable
signal quality over days, we unfortunately had to prioritize repetitions over stimulus set size.

Using a small number of repeated images also makes it highly likely that the visual system could
have learned about the statistics of those particular images, and this makes it harder to make
conclusions about how the system would respond to unpredictable images. It would have been
useful to have a condition where random images (selected from the ImageNet database for
example) were shown. Would such images produce more (or perhaps less) neural activity than the
highly predictable images used in the study? Would the authors predict that only the familiar stimuli
would show interesting sequences? If they authors have any such data, it would greatly enhance the
value of the data. My bet is that even a totally novel natural image would show clear and reliable
sequences of activation. It might even show better reliability than scrambled images (as
demonstrated here) but with no need for learning at all.

Previous work from our lab (Lazar et al., PNAS 2021; Peter et al., eLife 2021) examined the effect of
learning through repetition or exposure. However, in the present study we did not require the
animals to learn about specific images at short time scales (minutes to hours). The priors of natural
scene statistics (e.g. Gestalt principles) should have already been embedded in the cortical networks
through evolution and postnatal experience. Thus, unfamiliar stimuli (e.g. random novel images
from ImageNet), as long as they comprise natural scenes and objects, should exhibit these
fundamental statistics. Thus, in full agreement with the reviewer's prediction, familiar but also novel
stimuli that match internal priors of natural scene statistics should produce more reliable and better
decodable sequences than their scrambled versions. We also agree with the reviewer that any image
can trivially produce sequential responses; it is the relationship between sequence decodability (or
the "quality" of sequences) and the goodness of match between stimuli and internal priors that is
particularly interesting for us. We should have emphasized this point.

Indeed over the course of the experiment the visual system may well have learned new statistics and
installed additional priors about the repeated images, but such learning should also occur for
morphed and scrambled images, and has hopefully been factored out when we compared between
different groups of stimuli. However, it is of great interest to investigate more systematically how
learning new visual priors depends on existing priors (Tse et al., Science, 2011 & 2007). We have in
another very comprehensive simulation study (Effenberger et al, in prep) examined the effect of

initially installing very general Gestalt priors on later learning about specific images. We found that
later learning profited substantially from being able to build on basic Gestalt priors.

The use of just 3 stimuli also markedly reduces the power of the analysis. The decoder never has to
choose between more than 3 stimuli and makes judgment in favor of one of the stimuli even when
nothing is present. I would have preferred an analysis where the system had to initiate a three-
choice response when there was enough information to conclude that a stimulus had been
presented, and not simply choose which of the three stimuli is the most likely at every instant.

It is unfortunately beyond our capacity to develop novel decoding algorithms that would yield
quantitative comparisons between several stimuli. Therefore we decided to apply well established
machine learning classifiers that are also widely used in neuroscience and that served our basic
purpose of discriminating among stimuli. Using more stimuli would probably still run into similar
issues since the classification algorithm is independent of the number of classes to be classified, as
long as sufficient data are collected for training.

In the modelling study, I was puzzled by the statement that “Already at the beginning of training the
networks produced sequences but that they were highly variable”. What would cause that
variability? Was noise added to the activity? I would have thought that even the simplest neural
model of center-surround receptive fields in the retina would have latencies that would vary
depending on the stimulus because higher local contrast lead to shorter latencies.

Yes, background noise was added to the network (L868). Here we wanted to make two points: (1)
learning reduces sequence variability, which might explain better decodability; (2) the network can
reproduce temporally scalable sequences in response to linearly ramping input.

I was also surprised that the modelling was done with a relatively boring image of a digit. Why not
use the natural images used for the study? And was the modelling done with a spiking neural
network simulator? Or a conventional network that represents firing rates with floating point
numbers? These are important issues that are currently unclear from the presentation.

We decided to describe technical details of the network in the Method section to avoid too much
distraction on reading the main text. The spiking network model (L340 in Results, L859 in Methods)
used Izhikevich neurons (L861 in Methods, all parameters included) and was simulated in custom-
written MATLAB codes (L860, now also stated more explicitly). The network is not a firing rate
model; otherwise, it would have been difficult to show raw spiking activity in Figure 4.

Indeed the modelling could have been more elaborate, but it was intended as a proof of concept so
we took one of the most well-known stimulus sets (MNIST) as input.

L340: To explore the possibility that learning stabilizes temporal sequences in a stimulus-specific
way, we trained a spiking neural network in a non-supervised way to acquire information about the
shape of a digit (Figure 4. Methods).

L859: We simulated a recurrent spiking neural network of $N = 250$ neurons where the ratio between
excitatory and inhibitory neurons was 4:1 ($N_e = 200$, $N_i = 50$, custom-written simulation in MATLAB).
We used the Izhikevich model to simulate single neurons, whose parameters were the same as in

(Izhikevich, 2003). The excitatory cells were modelled as regular spiking cells (RS. $(a, b) = (0.02, 0.2)$.
$(c, d) = (-65, 8) + (15, -6) \cdot r^2$ where r is random variable of standard uniform distribution) and
inhibitory neurons as fast spiking cells (FS. $(a, b) = (0.02, 0.25) + (0.08, -0.05) \cdot r^2$. $(c, d) = (-65, 2)$),
each with random heterogeneity. All neurons were randomly connected with a connection
probability of 40%. The connection strength was initialized to be uniformly distributed between $[0,$
$1]$. To simulate external input current, we used MNIST digit images scaled to the size of 7-by-7, and
mapped to a subset of 49 excitatory neurons. In addition, all neurons received random background
noise as external input.

The paper refers to values for the total spikes per second that are several thousands. This is a
strange way to talk about the data, and is almost meaningless, because it is presumably calculated
for an undefined and large number of channels and where is it unclear whether the channels refer to
individual neurons or multiunit activity.

We used “total spikes per second” to measure *population* firing rate, i.e., summed firing rate
amplitudes across *all* channels, in the context of referring to the level of overall activity. Following
the reviewer’s suggestion, we have averaged total spike counts across channels as is usual in
neurophysiological studies and converted the values to “spikes per second”. All associated
quantifications have also been re-calculated.

Figure 1b:

L209 (caption of Figure 1): Average firing rates for different ramp conditions. Raster plots of single-
trial MUA responses to one of the three stimuli in the no ramp (left), fast ramp (middle), and
slow ramp (right) condition. The colored traces show the average firing rate per channel for each ramp
condition. Shaded areas denote 95% confidence level.

L135: The ramping stimuli reduced the peak amplitudes of the transient response components (no
ramp 51.57 ± 0.44 spikes/s, s.e.m.; fast ramp 48.59 ± 0.45 spikes/s; slow ramp 45.11 ± 0.45 spikes/s,
corresponding to reductions of 5.8% and 12.5%, respectively; all stimulus and intensity conditions
combined, $F_{2,927} = 52.79$, $p < 0.01$, one-way ANOVA; all pair-wise comparisons $p < 0.01$).

Figure 5 a&b:

L415 (caption of Figure 5): Lower panels: corresponding average firing rates per channel.

L390: Stimulus-specific information reached its maximum at the same time as the firing rate

(average per channel, Figure 5 a & b).

Supplementary Figure 15:

The authors also spend a lot of time discussing the information about peak latency for different
 channels. Indeed, much of the data in support of reliable sequences is built on the assumption that
 information about the peak latency in different channels could be used in the brain. It may be trivial
 for the experimenter to measure the latency of the peak. But firstly, this makes the basic assumption
 that rates are critical, whereas the whole point of rank order coding is to get away from rates.
 Secondly, I know of no plausible neurophysiological mechanism that could respond selectively to a
 peak at a particular latency. In my opinion, all that section could be dropped with no real loss. Onset
 latency variations across populations of neurons are by contrast intrinsically much more interesting,
 as well as being much faster to compute – as the authors point out.

We agree with the reviewer that onset latency is much more interesting and physiologically relevant.
 We initially used peak latency as the measure of response timing to not deviate too radically from
 conventions in the field, where neuronal response sequences are usually presented in heat maps of
 firing rates sorted by peak amplitude positions. We fully agree with the reviewer and are aware of
 the ambiguities inherent in decoding peak latencies. This is the reason why we designed a method
 that allowed us to determine precise onset latencies. We would like to also present peak latencies,

even if this is redundant, because we anticipate readers familiar with conventional representations
would ask for it.

So, in conclusion, I must admit that while interesting, the paper reaches conclusions that are not
supported by the data.

We hope that we were able to demonstrate with our replies to the reviewer's concerns and with our
revisions of the manuscript that the conclusions as they are formulated now are supported by the
data and we wish to thank again the reviewer for helping us to identify ambiguities and to improve
our writing.

REVIEWER COMMENTS

Reviewer #1 (Remarks to the Author):

The authors have addressed my concerns satisfactorily. This work should be published and shared with the community.

Reviewer #2 (Remarks to the Author):

The authors have worked hard to deal with the various points I raised in my review and I think that the paper is very much improved as a result.

Most of the points were dealt with well, and many of the questions have been answered in both the rebuttal and the modified manuscript.

That said, I still feel that the authors are overselling the strength of their evidence for recurrent mechanisms, although they have added many caveats which make their statements safer.

In the new version, they state that "to the best of our knowledge, there are yet no experimental data in support of feedforward mechanisms capable of generating response sequences, whose sequence order is invariant to temporal compression and dilation."

That may be the case, but only because almost no experimentalists have looked at the question! The fact is that even the simplest feedforward models would have to make that prediction. If you have a set of retinal ganglion cells responding to a flashed image, you would expect them to fire in a particular order that depended on how well the local image matched the receptive field – as shown by Gollisch and Meister, for example. True, those studies didn't look at what would happen if you changed the contrast, or ramped the image on slowly. But I would expect that the ordering information would be preserved, even in the face of such variations. Indeed, in 1998, Thorpe and Gautrais's proposals on Rank Order Coding took advantage of this by pointing out that the order in which the cells fired would remain the same despite wide variations in contrast and overall luminance. This is very close to the situation described here.

But we are talking about models - not experimental data.

So, no, I don't accept the authors' view that "recurrent processing as the most parsimonious explanation of response sequences and rank order codes". They argue that "Any recurrent network can be unrolled into a multilayer feedforward network and with appropriate implementation of nonlinearities, such feedforward networks can in principle simulate recurrent processes." This is one way of making feedforward models sound excessively complicated and hence tipping the balance in favour of recurrent mechanisms.

But I don't accept that you need multilayer networks with non-linearities to get sequences. Even a single layer of retinal ganglion cells will generate sequences in response to the onset of an image. You need integrate and fire, but nothing fancy. There is no need for anything else. And you would have to expect that the ordering in those sequences would remain stable even if you shifted contrast and luminance up and down.

All this is rather sad. The authors are to be commended on the fact that, at long last, there are experiments that have looked at these fascinating phenomena. And I really think that the data needs to be published. But I think that they really need to tone down even more their claims that sequences mean recurrent processing.

On the other points I made, I was happy to see that the authors accepted that there were a lot of limitations with the way the experiments were done. I think they know very well that the dataset would be far more interesting if they had included

- Single unit data rather than multiunit data
- Lots more images to avoid the criticism that 3 stimuli are not enough
- Track learning of a totally new image to see whether learning is needed for sequences (I don't believe it is).

I can't wait for the next dataset that deals with these limitations!

For the time being, I will just say that I would vote for publication of the current data with more caveats and the acceptance that sequences can be generated in a single layer network with no recurrent connections and nothing fancy in terms of non-linearities.

**REVIEWER COMMENTS (black colour)**

Reviewer #2 (Remarks to the Author):

The authors have worked hard to deal with the various points I raised in my review and I think that
the paper is very much improved as a result.

Most of the points were dealt with well, and many of the questions have been answered in both the
rebuttal and the modified manuscript.

That said, I still feel that the authors are overselling the strength of their evidence for recurrent
mechanisms, although they have added many caveats which make their statements safer.

**AUTHOR RESPONSES (green colour)**

We thank the reviewer very much for the re-review of our paper. Following the reviewer's
suggestion, we have further tuned down our interpretations, stating explicitly, that we do not wish
to exclude that sequences could also be generated in feedforward networks. In addition, we added a
reference to a recently published very extensive simulation study (Effenberger *et al.*, 2022) of a
cortical network that might be of interest for our readers. Deleted sentences are marked in
magenta. Added or modified sentences are marked in yellow.

As the reviewer had no further criticism of the results of our paper and voted for publication, we
hope, that our revisions are now to the satisfaction of the reviewer.

In the new version, they state that "to the best of our knowledge, there are yet no experimental
data in support of feedforward mechanisms capable of generating response sequences, whose
sequence order is invariant to temporal compression and dilation."

We have deleted this sentence.

That may be the case, but only because almost no experimentalists have looked at the question! The
fact is that even the simplest feedforward models would have to make that prediction. If you have a
set of retinal ganglion cells responding to a flashed image, you would expect them to fire in a
particular order that depended on how well the local image matched the receptive field – as shown
by Gollisch and Meister, for example. True, those studies didn't look at what would happen if you
changed the contrast, or ramped the image on slowly. But I would expect that the ordering
information would be preserved, even in the face of such variations. Indeed, in 1998, Thorpe and
Gautrais's proposals on Rank Order Coding took advantage of this by pointing out that the order in
which the cells fired would remain the same despite wide variations in contrast and overall
luminance. This is very close to the situation described here.

We added a paragraph in the discussion under "Methodological considerations" where we indicated
the shortcomings of our approach to distinguish between feedforward and recurrent processing and
propose experiments to accomplish this distinction. Here we now also mention as a methodological
limitation that we used only few stimuli and recorded only multi-unit activity.

But we are talking about models - not experimental data.

So, no, I don't accept the authors' view that "recurrent processing as the most parsimonious
explanation of response sequences and rank order codes". They argue that "Any recurrent network
can be unrolled into a multilayer feedforward network and with appropriate implementation of
nonlinearities, such feedforward networks can in principle simulate recurrent processes." This is one

way of making feedforward models sound excessively complicated and hence tipping the balance in
favour of recurrent mechanisms.

But I don't accept that you need multilayer networks with non-linearities to get sequences. Even a
single layer of retinal ganglion cells will generate sequences in response to the onset of an image.
You need integrate and fire, but nothing fancy. There is no need for anything else. And you would
have to expect that the ordering in those sequences would remain stable even if you shifted contrast
and luminance up and down.

All this is rather sad. The authors are to be commended on the fact that, at long last, there are
experiments that have looked at these fascinating phenomena. And I really think that the data needs
to be published. But I think that they really need to tone down even more their claims that
sequences mean recurrent processing.

We have now tuned down our claims at numerous places throughout the manuscript and
emphasize, that we cannot distinguish between feedforward and recurrent mechanisms (These
passages are marked in yellow). However, since the main motivation for the study was our previous
work on recurrent processing, we had to formulate our working hypothesis in the introduction. This
hypothesis was based on predictions derived from recurrent processing. Therefore, it was
unavoidable to discuss, whether our results are or are not compatible with our initial hypothesis.
Otherwise we would have betrayed our motivation to initiate this study.

On the other points I made, I was happy to see that the authors accepted that there were a lot of
limitations with the way the experiments were done. I think they know very well that the dataset
would be far more interesting if they had included

- Single unit data rather than multiunit data

- Lots more images to avoid the criticism that 3 stimuli are not enough

These two limitations have now been mentioned (see above).

- Track learning of a totally new image to see whether learning is needed for sequences (I don't
believe it is).

This is correct. Recurrent networks always produce sequences. We now emphasize that our
simulated network already generated sequences in the naive state, and that these sequences
became refined by learning, which improved their decodability.

For the time being, I will just say that I would vote for publication of the current data with more
caveats and the acceptance that sequences can be generated in a single layer network with no
recurrent connections and nothing fancy in terms of non-linearities.

We deleted the reference to nonlinearities and hope to have now added sufficient caveats.

REVIEWERS' COMMENTS

Reviewer #2 (Remarks to the Author):

I think that the additional caveats mean that the authors are no longer making unsubstantiated claims, and I am therefore happy to recommend publication.

I very much look forward to seeing further work on this important topic using protocols that go beyond the limitations of the current experiments.